# Clinical characteristics associated with mortality of COVID-19 patients admitted to an intensive care unit of a tertiary hospital in South Africa

**Peter S. Nyasulu**[1]*, **Birhanu T. Ayele**[1], **Coenraad F. Koegelenberg**[2], **Elvis Irusen**[2], **Usha Lalla**[2], **Razeen Davids**[3], **Yazied Chothia**[3], **Francois Retief**[4], **Marianne Johnson**[4], **Stephen Venter**[4], **Renilda Pillay**[4], **Hans Prozesky**[5], **Jantjie Taljaard**[5], **Arifa Parker**[5], **Eric H. Decloedt**[6], **Portia Jordan**[7], **Sa'ad Lahri**[8], **M Rafique Moosa**[9], **Muhammad Saadiq Moolla**[9], **Anteneh Yalew**[1], **Nicola Baines**[1], **Padi Maud**[2], **Elizabeth Louw**[2], **Andre Nortje**[2], **Rory Dunbar**[10], **Lovemore N. Sigwadhi**[1], **Veranyuy D. Ngah**[1], **Jacques L. Tamuzi**[1], **Annalise Zemlin**[11], **Zivanai Chapanduka**[12], **René English**[13], **Brian W. Allwood**[2]

1 Division of Epidemiology and Biostatistics, Department of Global Health, Faculty of Medicine and Health Sciences, Stellenbosch University, Cape Town, South Africa, 2 Division of Pulmonology, Department of Medicine, Faculty of Medicine and Health Sciences, Stellenbosch University & Tygerberg Hospital, Cape Town, South Africa, 3 Division of Nephrology, Department of Medicine, Faculty of Medicine and Health Sciences, Stellenbosch University & Tygerberg Hospital, Cape Town, South Africa, 4 Department of Anesthesia and Critical Care, Faculty of Medicine and Health Sciences, Stellenbosch University & Tygerberg Hospital, Cape Town, South Africa, 5 Division of Infectious Diseases, Department of Medicine, Faculty of Medicine and Health Sciences, Stellenbosch University & Tygerberg Hospital, Cape Town, South Africa, 6 Division of Clinical Pharmacology, Department of Medicine, Faculty of Medicine and Health Sciences, Stellenbosch University & Tygerberg Hospital, Cape Town, South Africa, 7 Department of Nursing, Faculty of Medicine and Health Sciences, Stellenbosch University, Cape Town, South Africa, 8 Department of Emergency Medicine, Faculty of Medicine and Health Sciences, Stellenbosch University & Tygerberg Hospital, Cape Town, South Africa, 9 Department of Medicine, Faculty of Medicine and Health Sciences, Stellenbosch University & Tygerberg Hospital, Cape Town, South Africa, 10 Department of Paediatrics & Child Health, Desmond Tutu TB Centre, Faculty of Medicine and Health Sciences, Stellenbosch University, Cape Town, South Africa, 11 Division of Chemical Pathology, Department of Pathology, Faculty of Medicine and Health Sciences, Stellenbosch University & NHLS Tygerberg Hospital, Cape Town, South Africa, 12 Division of Haematological Pathology, Department of Pathology, Faculty of Medicine and Health Sciences, Stellenbosch University & NHLS Tygerberg Hospital, Cape Town, South Africa, 13 Division of Health Systems and Public Health, Department of Global Health, Faculty of Medicine and Health Sciences, Stellenbosch University, Cape Town, South Africa

* pnyasulu@sun.ac.za

## Abstract

### Background

Over 130 million people have been diagnosed with Coronavirus disease 2019 (COVID-19), and more than one million fatalities have been reported worldwide. South Africa is unique in having a quadruple disease burden of type 2 diabetes, hypertension, human immunodeficiency virus (HIV) and tuberculosis, making COVID-19-related mortality of particular interest in the country. The aim of this study was to investigate the clinical characteristics and associated mortality of COVID-19 patients admitted to an intensive care unit (ICU) in a South African setting.

**Data Availability Statement:** All relevant data are within the paper and its Supporting Information files.

**Funding:** The author received rapid covid support grant from the Vice Rectors Office of Stellenbosch University to support with data collection, data entry and statistical analysis. An amount of R100,000 was disbursed in May 2020, and we used the funds to support data collection, development of REDCap data base, data entry, data quality control and statistical analysis. This work was mostly done by our PhD and masters students with oversight of the Lead investigator. The Office of the Vice Rector had no role in the conduct of the research nor in the drafting of this manuscript. The funder had no role in study design, data collection and analysis, decision to publish, or preparation of the manuscript.

**Competing interests:** The authors have declared that no competing interests exist.

## Methods and findings

We performed a prospective observational study of patients with severe acute respiratory syndrome coronavirus 2 (SARS-CoV-2) infection admitted to the ICU of a South African tertiary hospital in Cape Town. The mortality and discharge rates were the primary outcomes. Demographic, clinical and laboratory data were analysed, and multivariable robust Poisson regression model was used to identify risk factors for mortality. Furthermore, Cox proportional hazards regression model was performed to assess the association between time to death and the predictor variables. Factors associated with death (time to death) at p-value < 0.05 were considered statistically significant. Of the 402 patients admitted to the ICU, 250 (62%) died, and another 12 (3%) died in the hospital after being discharged from the ICU. The median age of the study population was 54.1 years (IQR: 46.0–61.6). The mortality rate among those who were intubated was significantly higher at 201/221 (91%). After adjusting for confounding, multivariable robust Poisson regression analysis revealed that age more than 48 years, requiring invasive mechanical ventilation, HIV status, procalcitonin (PCT), Troponin T, Aspartate Aminotransferase (AST), and a low pH on admission all significantly predicted mortality. Three main risk factors predictive of mortality were identified in the analysis using Cox regression Cox proportional hazards regression model. HIV positive status, myalgia, and intubated in the ICU were identified as independent prognostic factors.

## Conclusions

In this study, the mortality rate in COVID-19 patients admitted to the ICU was high. Older age, the need for invasive mechanical ventilation, HIV status, and metabolic acidosis were found to be significant predictors of mortality in patients admitted to the ICU.

## 1. Background

The outbreak of severe acute respiratory syndrome coronavirus 2 (SARS-CoV-2) infection causing Coronavirus disease 2019 (COVID-19) began in Wuhan, China in December 2019 and was declared a pandemic by the World Health Organization (WHO) on the 11th of February 2020 [1]. Severe cases of COVID-19 can present with acute respiratory distress syndrome (ARDS), acute kidney injury (AKI), acute cardiac injury (ACI) and sometimes, sudden unexplained death [2, 3]. Although SARS-CoV-2 susceptibility is universal, older age has always been associated with disease severity and high mortality [4].

As of the 07 November 2022, there have been over 629 million confirmed cases of COVID-19 and 6.5 million deaths reported globally since the start of the pandemic [5]. In the WHO Africa region, the number of confirmed cases was about 9.3 million, with about 174 thousand cumulative deaths [5]. During this period, a total of 4.0 million cases were reported in South Africa, of which 702, 220 cases were in the Western Cape Province and 101,982 deaths in the country [6].

Critically ill COVID-19 patients often require admission to the intensive care unit (ICU) for respiratory support, including invasive mechanical ventilation (IMV), non-invasive mechanical ventilation (NIV) and high flow nasal canula (HFNC) oxygen therapy [3]. ICU resources for the management of critically ill patients in low-and-middle-income countries (LMIC) especially in Sub-Saharan Africa (SSA) are limited compared to those in high-income countries (HIC) [7, 8].

According to the literature, the most common complications of COVID-19-related ARDS are AKI, disseminated intravascular coagulation, hepatic injury, and cardiac injury, which include myocarditis, pericarditis, pericardial effusion, arrhythmia, septic shock, sudden cardiac death [9] and encephalitis [10]. In a retrospective study of 1591 COVID-19 patients admitted to ICU in Italy, 26% [95%CI, 23%-28%] had died in the ICU at the time of this report with older patients (n = 786; age $\geq$ 64years) having a higher mortality rate than the younger patients (n = 795; age $\geq$ 63 years), (36% vs 15%; difference, 21% [95% CI, 17%-26%]; $P < .001$) [11]. A recent study conducted in China reported a case fatality rate of 53.8% (95% CI 50.1%-57.4%) by day 28 of ICU admission [12]. Another report from the USA reported a mortality rate of 291/371(78%) [13]. Diabetes mellitus, obesity, systemic hypertension, and chronic lung disease were the most frequently identified comorbidities [11–13]. The primary treatments for COVID-19-induced ARDS are oxygen therapy and respiratory support. The use of HFNC oxygen therapy upon ICU admission in adult patients with COVID-19-related ARDS may result in an increase in ventilator-free days and a reduction in ICU length of stay [14].

South Africa differs from other SSA countries in having a quadruple burden of disease with a higher prevalence of non-communicable diseases (12% for type 2 diabetes and 35% for hypertension) when compared to many other SSA countries [15]. In addition, the country is also contending with the so-called "colliding epidemics" of human immunodeficiency virus (HIV), tuberculosis (TB) (both active and post-TB lung disease) and chronic obstructive pulmonary disease (COPD) [16]. According to 2018 estimates, South Africa bears the double burden of HIV and tuberculosis infectious disease epidemics, with 7.7 million people living with HIV/ acquired immunodeficiency syndrome (AIDS) and 301,000 TB cases per year [17, 18]. Recent reviews have shown that both HIV and TB are associated with a higher risk of mortality from COVID-19 [19, 20].

The worsening COVID-19 pandemic in South Africa presents clinical decision-making challenges in the context of already-scarce ICU resources [21]. ARDS affects approximately 20% of hospitalized patients with confirmed COVID-19 pneumonia, with 12% requiring intubation and IMV [13, 22, 23]. The capacity in South Africa to treat a predicted number of patients with acute hypoxaemic respiratory failure (AHRF) with mechanical ventilation in the ICU is severely constrained [23, 24].

Previous studies have shown that the case fatality rates (CFR) of COVID-19 are higher with comorbidities such as diabetes and hypertension [22–25]; however, there is little known about clinical outcomes of COVID-19 patients with HIV, TB, post-TB lung disease (PTLD), rheumatic diseases, diabetes, and hypertension [4, 26]. The aim of the study was to document the mortality and associated demographic, clinical and laboratory characteristics, of COVID-19 patients admitted to a dedicated ICU in Cape Town, South Africa.

## 2. Methods

### 2.1. Ethics statement

The study was approved by the Health Research Ethics Committee (HREC) of Stellenbosch University's Faculty of Medicine & Health Sciences (N20/04/002 COVID-19). This ethical body granted the investigators a consent waiver.

### 2.2. Study population

The study was conducted at Tygerberg Hospital, a 1380-bed tertiary hospital in the East Metropole of Cape Town. The hospital provides tertiary services to approximately 3.5 million people from the Western Cape Province. The capacity in the ICU varied during this period between 17 to 44 beds. Critical care services were quadrupled in anticipation of the need [25].

Much of the population served by the hospital is from low-income areas, with a considerable proportion living in low-cost and informal settlements, where overcrowding, shared ablution and shared water facilities make social distancing and the advocacy of preventative hygiene methods difficult. The study population comprised all consecutive patients admitted to the adult ICU (age ≥ 18 years) between 27 March 2020 when the first patient was admitted, until 4 November 2020, when the database was censored from the first wave of COVID. Patients referred to the ICU were triaged by the consultants on duty according to disease severity and likely prognosis, according to Critical Care Society of Southern Africa (CCSSA) guidelines, and admissions depended on ICU bed availability [27]. The initial assessment of the referred patient is focused on determining whether the patient is critically ill and requires ICU admission for ventilatory support or other organ support only available in the ICU [27]. During the first wave all ventilatory support (invasive and non-invasive) were provided in a dedicated COVID-19 ICU [27].

## 2.3. Case management

Due to severely limited resources, the ICU management instituted a policy of initiating HFNC oxygen therapy for respiratory support in most admitted patients. The decision to intubate for mechanical ventilation was left to the discretion of the attending clinicians and was made on a case-by-case basis. There was very limited availability and utilization of specific antiviral therapies (e.g., remdesivir), but corticosteroids and therapeutic dose anti-coagulation were a unit policy if there were no contra-indications. During the pandemic's initial phase, all patients admitted were commenced on broad-spectrum antibiotics, while awaiting COVID-19 PCR results. Later, antibiotics were only administered if there was a clear suspicion of a secondary bacterial or nosocomial infection.

## 2.4. Data collection

Data were captured prospectively daily using photographs of clinical notes at the bedside, which were securely stored electronically, and clinical data were entered remotely by data-capturers into a REDCap® database; laboratory results were imported into the database. Data were checked by the 'data entry supervisor'.

## 2.5. Outcomes and predictor variables

Data collected included demographic and lifestyle characteristics data (age, sex, smoking status, alcohol use), clinical disease characteristics, pre-existing comorbidities (hypertension, diabetes, cardiovascular disease (CVD), chronic lung disease, obesity, and chronic kidney disease (CKD)), arterial blood gasses (pH, PaCO2, PaO2, potassium, lactate, bicarbonate, oxygen saturation, PF ratio), routinely collected laboratory data, ventilator support and oxygen requirements. A normal eGFR is 60 or more. A pH above 7.44 was considered as alkalemia. The primary outcome of interest was the proportion of patients who died after admission to the ICU including those who were discharged from the ICU but died in-hospital. Time to death or censored (alive at discharge) and length of stay in ICU was captured as per recorded documents.

## 2.6. Statistical analysis

Continuous variables were expressed as the mean with standard deviation (SD) for normally distributed data and the median with inter-quartile range (IQR) for non-normal data. Categorical variables were expressed using frequencies and percentages. Chi-square and Wilcoxon-

ranksum tests were performed to test the population distribution associated with mortality among categorical variables and the difference in medians for continuous variables with p-values. A robust Poisson regression model was used to assess significant associations between demographic, clinical factors, and death. Factors associated with death at a p-value < 0.15 in unadjusted univariable robust Poisson regression were included in a multivariable model, to identify independent factors associated with death. Due to the high prevalence of mortality, the logistic regression was overestimating the effect measure with large standard errors resulting in wide confidence intervals. Therefore, a robust Poisson regression model was used. Adjusted incidence rate ratios and their 95% CIs were used as a measure of association. Kaplan-Meier plots and log-rank tests were used to assess the association between time to death and the predictors. Furthermore, the predictors of mortality were assessed using the standard Cox proportional hazards model, incorporating clinically important variables selected *a priori* for the model. Covariate effects were assumed to be constant over time and Schoenfeld residuals were used to assess violations of the proportional hazard assumptions. Factors associated with death (time to death) at a p-value < 0.05 were considered statistically significant. All statistical analyses were performed using Stata (V.16, Stata Corp, College Station, Texas, USA) and R (V, 4.0.2, R Core Team) with R Studio (V.1.3, R Studio Team) statistical software.

## 3. Results

In this cohort, 413 patients were admitted to the ICU from 27 March to 4 November 2020. Fig 1 describes the cascade of care given and the corresponding clinical outcomes.

Of those admitted to the ICU, 227 (55%) were males, the median age was 54.1 years (IQR: 46.0–61.6), and 95% lived in the Cape Town Metropolitan Area. Underlying comorbidities were obesity (67%), hypertension (60%), diabetes mellitus (51%), HIV (14%), hyperlipidaemia (11%), TB (7%), asthma (5%), chronic kidney disease (CKD) (5%), insulin resistance (4%),

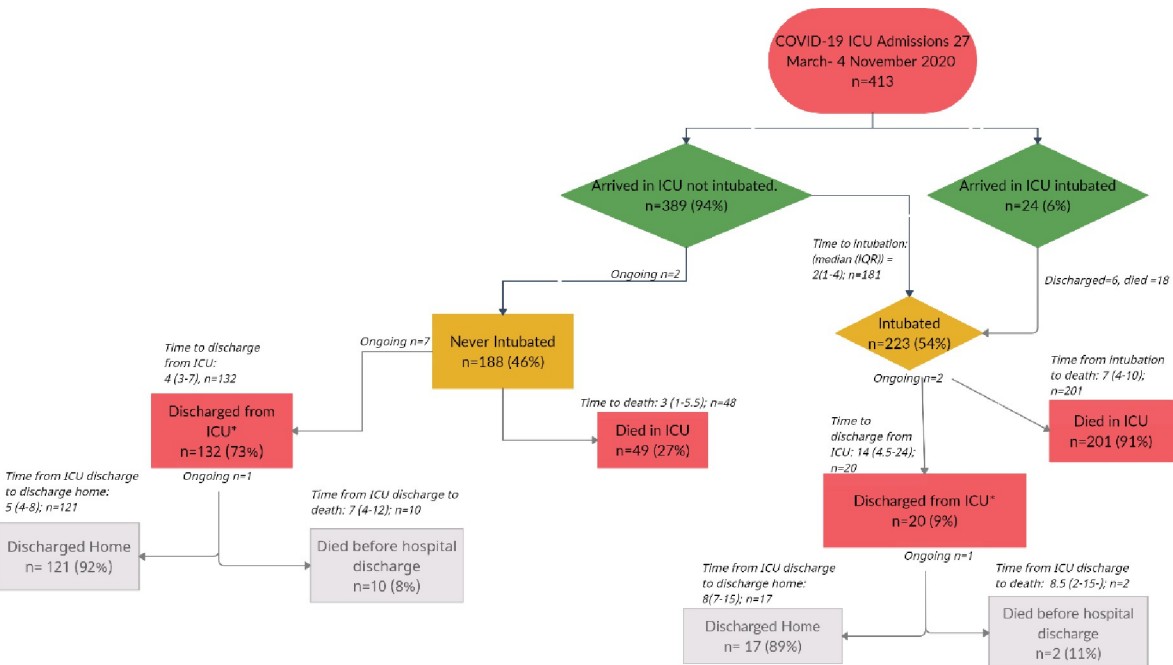

**Fig 1. Distribution and outcome of COVID-19 patients admitted in the ICU.**

ischaemic heart disease (IHD) (3%) and chronic obstructive pulmonary disease (COPD) (3%) [Table 1].

The median duration of stay in the ICU was 6 (IQR: 3–9) days, and for the 402 in whom outcome data was available at database censure, 262 (65%) died in hospital of whom 250 died in ICU (62%) [Table 1]. Clinical features at presentation to hospital included dyspnoea (91%), cough (84%), fever (49%), sore throat (23%), myalgia (22%), with the majority presenting with a tachycardia, but without hypotension or pyrexia [Table 1].

On admission to hospital, 79% of patients had an oxygen saturation < 90%, and arterial blood gases showed a median pH (IQR) of 7.47 (7.41–7.50), $PaO_2$ of 7.2 (6.0–9.1) kPa, median ratio of arterial oxygen partial pressure to fractional inspired oxygen (PF ratio) of 77.8 (54.4–116.0) [Table 1].

Most patients were initiated on HFNC (n = 389, 94%) on admission, which was consistent with the unit policy. Of these, 188 (46%) were never intubated [Fig 1] having an initial median PF ratio of 84.6 (IQR: 56.8–126.0); 132 (73%) of these were discharged from ICU after a median of 4 (IQR: 3–7) days and 49 (27%) died in ICU without being intubated and put on mechanical ventilation after a median of 3 days (IQR 1.0–5.5) [Fig 1].

The median initial PF ratio of those who ended up being intubated and mechanically ventilated (n = 223, 54%) was 73.5 (IQR: 52.5–106.7). Of these, 201 (91%) died after a median of 7 (IQR: 4–10) days, and 20 (9%) were discharged after a median of 14 (IQR: 4.5–24) days. The median PF ratio on admission was significantly higher for those who survived after being intubated compared with those who did not, being 93.7 (IQR: 63.7–141.4) and 70.3 (51.0–91.2), respectively (P<0.001).

The overall ICU mortality was 62% (250/402). Of those who were discharged from the ICU 92% (138/150) survived and 8% (12/150) died prior to hospital discharge. A total of 223 patients were intubated and ventilated; of these, 24 were intubated before admission to the ICU and 199 were intubated during their ICU stay. In total, of those who were intubated before ICU, 20 (83%) died compared to 181 (91%) who were intubated in the ICU (P = 0.416). The overall mortality among this cohort (of mechanically ventilated patients was 181/204 (88.7%). On the crude incidence rate ratio (CIRR), there were significant gender differences on discharge (30% females versus 44% males) and mortality (70% females versus 56% males), P = 0.005 (Tables 2 and 3). However, this difference was not statistically significant on adjusted IRR (aIRR) (P = 0.209, Tables 2 and 3). Age on admission was statistically different between the discharged patients and those who died, on both the CIRR and aIRR (P < 0.001, Tables 2 and 3). Patients with an oxygen saturation less than 90% on admission to the ICU were at higher risk of death as compared to those with more than 90% (CIRR = 1.44, 95% CI: 1.11–1.87, P = 0.007). Apart from HIV, none of the pre-existing comorbidities was significantly associated with mortality in the multivariate analysis, however, hypertension and type 1 and 2 diabetes mellitus were associated with higher mortality in univariate analysis. However, among HIV-infected patients, only (13.9%) 54/388 were on ART and tenofovir disoproxil fumarate (TDF)/emtricitabine (FTC)/efavirenz (EFV) was the prevalent regimen (43.54%).

In adjusted multivariable analysis, the mortality rate was associated with age (aIRR = 1.01, 95%CI: 1.01–1.02, P < 0.001), Myalgia (aIRR = 0.83, 95% CI: 0.71, 0.97, p < 0.021), intubation (aIRR = 2.57, 95%CI: 2.08–3.18, P <0.001 and HIV positive status (aIRR = 1.30, 95%CI: 1.06–1.59, P = 0.012) [Table 2]. In unadjusted multivariate analysis, other factors were associated with the mortality rate [Table 2].

In adjusted laboratory results multivariable analysis, the mortality rate was associated with PCT (aIRR = 1.0004, 95%CI: 1.0001–1.001, P = 0.002), Troponin T (aIRR = 1.002, 95%CI: 1.0003–1.002, P = 0.017 and AST (aIRR = 1.001, 95%CI: 1.0001–1.003, P = 0.037) [Table 3]. In

**Table 1. Frequency distribution of socio-demographic and bio-clinical characteristics of COVID-19 patients admitted in the ICU.**

| Characteristics | Sample size* | Number (%) or Median (IQR: Q1-Q3) |
|---|---|---|
| **Hospital** | | |
| Died in hospital | 400 | 262 (65) |
| Length of hospital stay (days) | 400 | 9 (6–14) |
| **ICU admission** | | |
| Time from hospital to ICU admission (days) | 411 | 1 (0–2) |
| Died in ICU | 402 | 250 (62) |
| Duration of ICU stay (days)** | 402 | 6 (3–9) |
| Intubated in ICU | 411 | 223 (54) |
| **Sociodemographic and lifestyle characteristics** | | |
| Age | 410 | 54.1 (46.0–61.6) |
| Male | 410 | 226 (55) |
| Cigarette smokers (past + current) | 202 | 48 (24) |
| Alcohol use | 147 | 16 (11) |
| **Clinical features at initial presentation** | | |
| Dyspnoea | 382 | 349 (91) |
| Cough | 382 | 323 (84) |
| Fever | 382 | 188 (49) |
| Sore throat | 382 | 87 (23) |
| Myalgia | 382 | 87 (22) |
| Headache | 382 | 50 (13) |
| Diarrhoea | 382 | 47 (12) |
| Acute kidney injury | 399 | 44 (11) |
| Vomiting | 382 | 15 (4) |
| Fatigue | 382 | 13 (3) |
| Proteinuria | 399 | 11 (3) |
| Oxygen saturation < 90% | 400 | 316 (79) |
| Arterial-blood gas—PaO2 | 399 | 7.1 (5.7–8.9) |
| BP—Systolic | 390 | 136 (120–152) |
| BP—Diastolic | 390 | 79 (70–88) |
| Temperature | 399 | 36.9 (36.3–37.7) |
| Glasgow coma scale (3–15) | 399 | 15 (15–15) |
| **Underlying comorbidities conditions** | | |
| BMI >30 kg/m$^2$ | 400 | 268 (67) |
| Hypertension | 400 | 239 (60) |
| Diabetes mellitus (type 1 or type 2) | 401 | 203 (51) |
| HIV positive | 394 | 55 (14) |
| Hyperlipidaemia | 400 | 40 (11) |
| Previous history of pulmonary TB | 388 | 28 (7) |
| Asthma | 400 | 21 (5) |
| Chronic kidney disease | 399 | 19 (5) |
| Ischaemic heart disease | 400 | 11 (3) |
| Post-TB lung disease | 399 | 12 (3) |
| **Arterial blood gasses** | | |
| pH | 395 | 7.47 (7.41–7.50) |
| PaCO$_2$(kPa) | 395 | 4.8 (4.3–5.5) |
| PaO$_2$ | 395 | 7.2 (6.0–9.1) |

(*Continued*)

**Table 1.** (Continued)

| Characteristics | Sample size* | Number (%) or Median (IQR: Q1-Q3) |
|---|---|---|
| Potassium (mmol/L) | 393 | 3.7 (3.3–4.2) |
| Lactate (mmol/L) | 387 | 1.4 (1.1–2) |
| Bicarbonate (mmol/L) | 375 | 26.6 (23.9–28.7) |
| Oxygen saturation | 384 | 89 (82–94) |
| PF ratio | 392 | 77.8 (54.4–116.0) |

*The purpose of presenting the sample size was to show the number of observations for each characteristic since each characteristic did not have an equal sample size.

**Days to death/discharge from ICU (0–51 days); HIV, human immunodeficiency virus; BMI, Body mass index; TB, tuberculosis; PaCO2, partial pressure of carbon dioxide; PaO2, partial pressure of oxygen; PF ratio, ratio of arterial oxygen partial pressure (in mmHg) to fractional inspired oxygen; IQR: interquartile range

unadjusted multivariable analysis, other factors were associated with the mortality rate [Table 3].

In the adjusted multivariable Cox regression analysis indicated that intubated patients in ICU were 1.58 times at higher risk of death than those who were not intubated (aHR 1.58, 95% CI: 1.10–2.25, P = 0.017). Similarly, HIV-infected patients had 1.64 times increased risk of dying compared HIV negative patients (aHR 1.59, 95%CI: 1.10–2.31, P = 0.015). In contrast, myalgia on admission was associated with 29% reduced risk of dying compared to those who did not present with this clinical symptom (aHR 0.70, 95%CI: 0.50–0.99, P = 0.049) [Table 4]. Furthermore, use of vancomycin, enoxaparin, proton pump inhibitors (PPI), spironolactone, losartan and other hypertensive drugs were associated with a low risk of mortality (aHR 0.62, 95%CI: 0.38–1.01, P = 0.011; aHR = 0.16, 95%CI: 0.06–0.46, P = 0.001; aHR 0.71, 95%CI: 0.53–0.97, P = 0.033; aHR = 0.50, 95%CI: 0.29–0.85, P = 0.011; aHR 0.58, 95%CI: 0.36–095, P = 0.03 and aHR 0.74, 95%CI: 0.55–0.99, P = 0.04 respectively) [Table 5]. Lastly, increased N-terminal pro-brain natriuretic peptide (NT-proBNP) was weakly associated with mortality (P = 0.039) [Table 6]. Lastly, the use of dexamethasone (daily dose 8mg), methyl prednisone (daily dose 40mg) and hydrocortisone (daily dose 200mg) were not with mortality with P-values of 0.429, 0.174 and 0.754, respectively [Table 6].

In terms of mortality progression over the study period as represented by the survival curves, the Kaplan-Meier product limit estimates indicated that the mortality rate remained high among intubated patients compared to non-intubated patients to about days 6–36 after admission [Fig 2].

Test of proportional-hazards assumption was performed for adjusted Cox proportional model, and it satisfied the assumption (p = 0.228) [Fig 3].

The study also found that high levels of N-terminal pro-brain natriuretic peptide (NT-proBNP) was associated with mortality aHR (95% CI) of 1.00 (1.00–1.0001) (P = 0.039) indicating the need for aggressive treatment and observation among such patients [Table 6].

## 4. Discussion

In this prospective cohort study of 413 patients with COVID-19 admitted to an ICU in Cape Town, South Africa, most patients were men (55%), and the median age was 54.1 years (IQR: 46.0–61.6). The overall mortality rate in ICU was 250/402 (62%). The mortality rate among those who were ventilated was much higher 201/221 (91%). In total, 20 (83%) of those intubated prior to ICU died, compared to 181 (91%) of those intubated in the ICU. The median

**Table 2. Socio-demographic and clinical characteristics associated with mortality among COVID-19 patients admitted to ICU.**

| Characteristics and level | Row %age | | | Mortality rate | | | |
| --- | --- | --- | --- | --- | --- | --- | --- |
| | | | | Crude IRR | | Adjusted IRR | |
| Forms and variables | Discharged Median (IQR) or n (%) | Died Median (IQR) or n (%) | p-value | CIRR (95% CI) | p-value | aIRR (95% CI) | p-value |
| Participants | 152 (38) | 250 (62) | | | | | |
| Intubated in ICU (221[1] of 401) | 20(9) | 201(91) | <0.001 | 3.41 (2.67–4.36) | <0.001 | 3.18 (2.46, 4.10) | <0.001 |
| Duration of pre-ICU admission[2] (250 of 402) | 1 (0–3) | 0 (0–2) | <0.001 | 0.94 (0.89–1.00) | 0.040 | | |
| **Sociodemographic and lifestyle characteristics** | | | | | | | |
| Age (250 of 402) | 50.4 (42.8–58.3) | 56.7 (48.0–63.1) | <0.001 | 1.02 (1.01–1.02) | <0.001 | 1.01(1.00, 1.02) | <0.001 |
| Male (224 of 402) | 98 (44) | 126 (56) | 0.006 | 0.81 (0.69–0.94) | 0.005 | 0.92 (0.80, 1.06) | 0.254 |
| Smoker (45 of 217) | 20 (44) | 25 (56) | 0.702 | 0.95 (0.71–1.27) | 0.708 | | |
| **Symptoms at initial presentation** | | | | | | | |
| Fever (185 of 374) | 77 (42) | 108 (58) | 0.150 | 0.89 (0.76–1.04) | 0.152 | | |
| Cough (315 of 374) | 122 (39) | 193 (61) | 0.483 | 0.92 (0.76–1.14) | 0.464 | | |
| Sore throat (86 of 374) | 41 (48) | 45 (52) | 0.035 | 0.81 (0.65–1.00) | 0.054 | 0.87 (0.73, 1.05) | 0.144 |
| Myalgia (86 of 374) | 40 (47) | 46 (53) | 0.063 | 0.83 (0.67–1.03) | 0.086 | 0.83 (0.71, 0.97) | 0.021 |
| Fatigue (12 of 374) | 5 (42) | 7 (58) | 0.788 | 0.94 (0.58–1.53) | 0.798 | | |
| Headache (49 of 374) | 22 (45) | 27 (55) | 0.284 | 0.87 (0.67–1.14) | 0.320 | | |
| Dyspnoea (341 of 374) | 128 (38) | 213 (62) | 0.417 | 1.12 (0.83–1.52) | 0.450 | | |
| Vomiting (14 of 374) | 4 (29) | 10 (71) | 0.460 | 1.16 (0.82–1.63) | 0.399 | | |
| Diarrhoea (44 of 374) | 20 (45) | 24 (55) | 0.276 | 0.87 (0.65–1.15) | 0.316 | | |
| Arterial-blood gas—PaO2 (240 of 391) | 7.6 (6.3–9.5) | 6.8 (5.2–8.3) | <0.001 | 1.00 (0.99–1.01) | 0.758 | | |
| BP–Systolic, (136 of 382) | 136 (121–156) | 136 (120–151) | 0.598 | 1.00 (0.99–1.00) | 0.467 | | |
| BP–Diastolic (235 of 382) | 79 (71–90) | 78 (68–87) | 0.088 | 1.00 (0.99–1.00) | 0.084 | 1.00 (0.99, 1.01) | |
| Pulse (235 of 383) | 105 (91–113) | 101(90–117) | 0.670 | 1.00 (0.99–1.00) | 0.768 | | |
| Temperature (240 of 391 | 36.8 (36.4–37.7) | 37 (36.1–37.6) | 0.591 | 1.00 (0.99–1.00) | 0.170 | | |
| Proteinuria (11 of 351) | 4 (36) | 7 (64) | 0.884 | 1.04 (0.66–1.63) | 0.881 | | |
| Acute respiratory distress syndrome (berlin) (72 of 375) | 31 (43) | 41 (57) | 0.456 | 0.92 (0.74–1.15) | 0.473 | | |
| Glasgow coma scale (241 of 392 | 15 (15–15) | 15 (15–15) | 0.052 | 1.00 (1.00–1.01) | 0.168 | | |
| Acute kidney injury (43 of 380) | 16 (37) | 27 (63) | 0.804 | 1.03 (0.81–1.32) | 0.800 | | |
| **Underlying comorbidities conditions** | | | | | | | |

*(Continued)*

**Table 2.** (Continued)

| Characteristics and level | Row %age | | | Mortality rate | | | |
|---|---|---|---|---|---|---|---|
| | | | | Crude IRR | | Adjusted IRR | |
| Forms and variables | Discharged Median (IQR) or n (%) | Died Median (IQR) or n (%) | p-value | CIRR (95% CI) | p-value | aIRR (95% CI) | p-value |
| Hypertension (233 of 292) | 81 (35) | 152 (65) | 0.084 | 1.15 (0.98–1.36) | 0.093 | 0.98 (0.86, 1.13) | 0.827 |
| Asthma (21 of 392) | 5 (24) | 16 (76) | 0.161 | 1.25 (0.97–1.61) | 0.083 | 1.18 (0.97, 1.45) | 0.104 |
| Diabetes mellitus (type 1 & 2) (197 of 393) | 66 (34) | 131 (66) | 0.056 | 1.16 (0.99–1.36) | 0.058 | 1.05 (0.92, 1.20) | 0.458 |
| Insulin resistance (17 of 391) | 9 (53) | 8 (47) | 0.206 | 0.76 (0.45–1.26) | 0.282 | | |
| Ischaemic heart disease (11 of 392) | 6 (55) | 5 (45) | 0.260 | 0.73 (0.38–1.40) | 0.346 | | |
| Hyperlipidaemia (42 of 392) | 13 (31) | 29 (69) | 0.302 | 1.13 (0.91–1.41) | 0.260 | | |
| Raised BMI (263 of 392) | 92 (35) | 171 (65) | 0.056 | 1.18 (0.99–1.41) | 0.069 | 1.15 (0.99, 1.34) | 0.061 |
| HIV positive (54 of 388) | 15 (28) | 39 (72) | 0.084 | 1.21 (1.00–1.45) | 0.050 | 1.24 (1.01, 1.51) | 0.040 |
| Previous history of pulmonary TB (28 of 380) | 10 (36) | 18 (64) | 0.782 | 1.04 (0.78–1.39) | 0.776 | | |
| Chronic lung disease (12 of 391) | 4 (33) | 8 (67) | 0.716 | 1.08 (0.72–1.63) | 0.697 | | |
| Chronic kidney disease (CKD) (18 of 391) | 5 (28) | 13 (72) | 0.344 | 1.18 (0.88–1.59) | 0.273 | | |
| **Arterial blood gasses** | | | | | | | |
| pH (238 of 385) | 7.47 (7.44–7.51) | 7.46 (7.40–7.49) | <0.001 | 0.27 (0.17–0.43) | <0.001 | 0.63 (0.32, 1.23) | 0.026 |
| paCO2 (238 of 385) | 4.7 (4.1–5.2) | 5.0 (4.3–5.7) | 0.006 | 1.04 (1.01–1.06) | 0.003 | | |
| paO2 (238 of 385) | 7.6 (6.3–9.6) | 6.9 (5.5–8.4) | <0.001 | 0.96 (0.93–0.99) | 0.006 | | |
| K+ (236 of 383) | 3.7 (3.4–4.1) | 3.8 (3.3–4.2) | 0.504 | 1.03 (0.92–1.15) | 0.579 | | |
| Lactate (231 of 377) | 1.4 (1.0–1.9) | 1.5 (1.1–2.1) | 0.064 | 1.04 (1.02–1.07) | 0.001 | | |
| HCO3std (225 of 368) | 26.9 (26.9–28.7) | 26.1 (23.9–28.6) | 0.176 | 1.00 (0.997–1.004) | 0.845 | | |
| So2c (229 of 374) | 91 (87–96) | 88 (79–92) | <0.001 | 0.99 (0.98–0.99) | <0.001 | | |
| PF ratio (235 of 382) | 93.8 (63.8–141.4) | 70.3 (51.0–91.3) | <0.001 | 1.00 (0.99–1.00) | 0.037 | 0.998 (0.995, 1.001) | 0.164 |

Data are n (%; row percentage) or median (IQR) for continuous variable and non-normal data in the descriptive table; CIRR: Crude incidence rate ratio; aIRR: adjusted for all other covariates in table; IRR: incidence rate ratio or risk ratio; CI: Confidence Interval. HIV: Human immunodeficiency virus; INR: International normalized ratio; PCT: procalcitonin; proBNP: pro B type natriuretic peptide; paCO2: Partial Pressure of Carbon Dioxide; paO2: Partial Pressure of Oxygen; pH: potential of Hydrogen; PPIs: Proton-pump inhibitors; PTT: Partial Thromboplastin Time.

** Variables with p-value less than 0.15 in the unadjusted PR, was kept in the final multivariable models and considered multicollinearity among predictors. As a result, we have excluded paCO2, paO2, Lactate and So2c in adjusted model due to multicollinearity even though p-value is <0.15 in an unadjusted model. [1]Number of deaths out of total patients

[2]Time to ICU from TBH admission (in days)

**Table 3. Laboratory results at initial measurements associated with mortality among COVID-19 patients admitted to ICU at Tygerberg Hospital.**

| Laboratory parameters | Row %age | | | Mortality rate | | | |
|---|---|---|---|---|---|---|---|
| | | | | Crude IRR | | Adjusted IRR | |
| | Discharged Median (IQR) or n (%) | Died Median (IQR) or n (%) | p-value* | CIRR (95% CI) | p-value | aIRR (95% CI) | p-value |
| Sodium (Dis = 152; D = 248) | 137 (134–139.5) | 137.5 (134–141) | 0.361 | 1.00 (0.99–1.02) | 0.407 | | |
| Creatinine (Dis = 152; D = 248) | 76.5 (62–99) | 77.5 (63–109.5) | 0.330 | 1.00 (0.999–1.001) | 0.748 | | |
| Haemoglobin (Dis = 152; D = 248) | 13.1 (12.35–14.2) | 13.1 (11.55–14) | 0.230 | 0.98 (0.94–1.02) | 0.345 | | |
| Haematocrit (Dis = 152; D = 248) | 0.402 (0.373–0.433) | 0.40 (0.359–0.433) | 0.497 | 0.84 (0.20–1.42) | 0.803 | | |
| Platelets (Dis = 152; D = 248) | 297.5 (216–383) | 288 (223.5–370.5) | 0.452 | 1.00 (0.999–1.000) | 0.432 | | |
| Urea (Dis = 152; D = 248) | 6.05 (4.4–8.65) | 6.6 (4.8–9.9) | 0.072 | 1.01 (1.001–1.02) | 0.028 | 1.00 (0.97–1.03) | 0.854 |
| White Cell Count (Dis = 152; D = 248) | 11.035 (7.675–13.365) | 11.605 (8.905–15.305) | 0.012 | 1.00 (0.99–1.01) | 0.783 | | |
| Lymphocytes (Dis = 152; D = 247) | 0.955 (0.655–1.325) | 0.94 (0.63–1.28) | 0.678 | 0.98 (0.86–1.11) | 0.766 | | |
| Neutrophils (Dis = 152; D = 247) | 8.58 (6.39–11.695) | 9.71 (7.35–13.32) | 0.004 | 1.02 (1.01–1.04) | 0.004 | 0.99 (0.96–1.03) | 0.750 |
| Chloride (Dis = 152; D = 246) | 99 (96–101) | 98 (95–102) | 0.748 | 1.00 (0.99–1.01) | 0.596 | | |
| CRP (Dis = 150; D = 245) | 173 (91–270) | 194 (125–297) | 0.019 | 1.00 (1.000–1.001) | 0.008 | 1.00 (0.99–1.00) | 0.259 |
| PCT (Dis = 151; D = 242) | 0.29 (0.13–0.94) | 0.55 (0.25–1.16) | <0.001 | 1.00 (1.0001–1.0003) | <0.001 | 1.0004 (1.0001–1.001) | 0.002 |
| D-Dimer (Dis = 148; D = 242) | 0.64 (0.39–2.49) | 1.33 (0.52–6.31) | <0.001 | 1.02 (1.01–1.03) | 0.04 | 1.00 (0.97–1.03) | 0.957 |
| Total bilirubin (Dis = 149; D = 235) | 8 (5–10) | 7 (5–10) | 0.207 | 0.997 (0.98–1.01) | 0.680 | | |
| Alanine Aminotransférase (ALT) (Dis = 142; D = 240) | 31 (20–55) | 31 (20.5–48) | 0.700 | 1.00 (1.00–1.00) | 0.910 | | |
| eGFR | | | 0.126 | | | | |
| Normal (n = 261), n (%) | 104 (40) | 157 (60) | | | | | |
| Non normal (n = 117), n (%) | 37 (32) | 80 (68) | | 1.14 (0.87–1.49) | 0.351 | | |
| proBNP (Dis = 137; D = 222) | 211 (47–738) | 485.5 (1482) | <0.001 | 1.00 (1.00–1.00) | 0.042 | 1.00 (1.00–100) | 0.984 |
| Ferritin (Dis = 138; D = 218) | 982 (589–1645) | 1131 (721–1795) | 0.212 | 1.00 (1.00–1.00) | 0.874 | | |
| INR (Dis = 138; D = 218) | 1.12 (1.04–1.20) | 1.13 (1.06–1.22) | 0.104 | 1.13 (0.97–1.30) | 0.114 | | |
| Troponin T (Dis = 140; D = 216) | 10.5 (6–19.5) | 18 (11–43.5) | <0.001 | 1.0003 (1.0001–1.001) | 0.029 | 1.002 (1.0003–1.002) | 0.017 |
| HbA1c (Dis = 114; D = 187) | 6.45 (6.1–8.1) | 6.9 (6.2–10) | 0.010 | 1.04 (1.01–1.07) | 0.012 | 1.05 (0.99–1.12) | 0.092 |
| Calcium (Dis = 88; D = 159) | 2.055 (1.97–1.16) | 2.05 (1.96–2.16) | 0.935 | 1.06 (0.63–1.78) | 0.840 | | |
| Magnesium (Dis = 87; D = 157) | 0.92 (0.80–1.96) | 0.94 (0.83–1.05) | 0.159 | 1.31 (1.05–1.63) | 0.016 | 1.30 (0.98–1.72) | |
| Phosphate (Dis = 86; D = 155) | 1.025 (0.85–1.17) | 0.99 (0.77–1.22) | 0.709 | 1.03 (0.82–1.28) | 0.813 | | |
| Aspartate Aminotransferase (AST) (Dis = 79; D = 123) | 44 (30–70) | 53 (38–75) | 0.078 | 1.0001 (1.0001–1.0002) | 0.007 | 1.001 (1.0001–1.003) | 0.037 |

Key: Data are n (%; row percentage) or median (IQR) for continuous non-normal data in the descriptive table; CIRR: Crude incidence rate ratio; aIRR: adjusted for all other covariates in table; IRR: incidence rate ratio or risk ratio; CI: Confidence Interval; Dis: discharge; D: death. Variables with number of observations less than 250 (rule of thumb) were not included in the adjusted model (e.g., AST). p-value* tests median equality

Abbreviations: ALP: alkaline phosphatase; ALT: Alanine Aminotransferase; AST: Aspartate Aminotransferase; CRP: C-reactive protein; GGT: Gamma-glutamyl transferase; eGFR: Glomerular Filtration rate; HbA1c: Haemoglobin A1C; HCO3-: bicarbonate; Hct: Haematocrit; INR: International normalized ratio; MCHC: Mean Corpuscular Haemoglobin Concentration; MCV: Mean Corpuscular Volume; MPV: Mean Platelet Volume; PCT: procalcitonin; proBNP: pro B type natriuretic peptide; paCO2: Partial Pressure of Carbon Dioxide; paO2: Partial Pressure of Oxygen; pH: potential of Hydrogen; PPIs: Proton-pump inhibitors; PTT: Partial Thromboplastin Time

ICU length of stay was 6 days (IQR: 3–9) days. The survival profiles showed that the mortality rate was higher among intubated than non-intubated patients.

Although the global mortality rate for COVID-19 patients on IMV is high, there seems to be significant variation between countries [28]. In another systematic review and meta-analysis of patients with severe COVID-19, the overall estimate for the reported mortality rate was 45%

**Table 4. Cox proportional hazards model identifying factors associated with the risk of hazard ratio among COVID-19 patients admitted to the ICU.**

| Characteristics | Hazard ratio | | | |
| --- | --- | --- | --- | --- |
| | Crude HR | | Adjusted HR | |
| | CHR (95% CI) | p-value | aHR (95% CI) | p-value |
| **ICU admission** | | | | |
| Intubated in ICU | 1.90 (1.37–2.64) | <0.001 | 1.58 (1.10–2.28) | 0.013 |
| Duration of pre-ICU admission[1] | 0.97 (0.90–1.04) | 0.354 | | |
| Sociodemographic and lifestyle characteristics | | | | |
| Age at admission | 1.01 (1.00–1.02) | 0.172 | | |
| Gender (male) | 0.88 (0.68–1.13) | 0.325 | | |
| Smoker | 0.88 (0.57–1.39) | 0.593 | | |
| **Clinical features at initial presentation** | | | | |
| Fever | 1.00 (0.77–1.30) | 0.997 | | |
| Cough | 1.09 (0.77–1.54) | 0.632 | | |
| Sore throat | 0.67 (0.48–0.94) | 0.021 | 0.70 (0.49–0.99) | 0.047 |
| Myalgia | 0.75 (0.54–1.03) | 0.076 | 0.71 (0.50–1.00) | 0.049 |
| Fatigue | 0.64 (0.30–1.36) | 0.245 | | |
| Headache | 0.94 (0.63–1.40) | 0.748 | | |
| Dyspnoea | 0.98 (0.62–1.56) | 0.946 | | |
| Vomiting | 1.52 (0.81–2.87) | 0.194 | | |
| **Vital Signs/other clinical parameters** | | | | |
| Blood pressure–systolic | 1.00 (0.99–1.00) | 0.366 | | |
| Blood pressure–diastolic | 0.99 (0.99–1.00) | 0.239 | | |
| Pulse | 1.00 (0.996–1.010) | 0.394 | | |
| Temperature | 0.99 (0.98–1.00) | 0.113 | 0.99 (0.97–1.01) | 0.161 |
| Acute respiratory distress syndrome (berlin definition) | 0.71 (0.60–1.01) | 0.054 | 0.70 (0.48–1.03) | 0.069 |
| Glasgow coma scale | 1.00 (0.99–1.01) | 0.776 | | |
| Severe/critical | 2.15 (1.12–4.05) | 0.019 | 1.59 (0.69–3.46) | 0.242 |
| Oxygen saturation < 90% | 1.43 (0.99–2.08) | 0.056 | 1.28 (0.83–1.99) | 0.264 |
| Acute kidney injury | 1.02 (0.68–1.54) | 0.911 | | |
| **Underlying comorbidities conditions** | | | | |
| Hypertension | 0.98 (0.75–1.27) | 0.852 | | |
| Asthma | 1.45 (0.87–2.42) | 0.151 | | |
| Diabetes mellitus | 1.11 (0.86–1.43) | 0.433 | | |
| Insulin resistance | 0.91 (0.45–1.84) | 0.790 | | |
| Ischaemic heart disease | 0.80 (0.3–1.94) | 0.617 | | |
| Hyperlipidaemia | 1.04 (0.70–1.53) | 0.847 | | |
| BMI > 30 kg/m$^2$ | 1.16 (0.88–1.54) | 0.298 | | |
| HIV positive | 1.44 (1.02–2.04) | 0.037 | 1.59 (1.10–2.31) | 0.015 |
| Previous history of pulmonary TB | 0.97 (0.60–1.58) | 0.911 | | |
| Chronic lung disease | 1.62 (0.80–3.29) | 0.180 | | |
| Chronic kidney disease | 0.90 (0.51–1.58) | 0.714 | | |
| **Arterial blood gasses** | | | | |
| pH | 0.17 (0.05–0.57) | 0.004 | 0.33 (0.08–2.31) | 0.129 |
| paCO$_2$ (kPa) | 1.01 (0.97–1.06) | 0.627 | | |
| paO$_2$ (kPa) | 0.97 (0.94–1.01) | 0.190 | | |
| Potassium (mmol/L) | 1.11 (0.92–1.36) | 0.255 | | |
| Lactate (mmol/L) | 1.08 (0.98–1.18) | 0.111 | | |
| HCO$_3$ (mmol/L) | 1.00 (0.99–1.01) | 0.798 | | |

*(Continued)*

**Table 4.** (Continued)

| Characteristics | Hazard ratio | | | |
|---|---|---|---|---|
| | Crude HR | | Adjusted HR | |
| | CHR (95% CI) | p-value | aHR (95% CI) | p-value |
| SO2 | 0.99 (0.98–1.00) | 0.042 | | |
| PF ratio | 1.00 (0.99–1.00) | 0.105 | 0.998 (0.995–1.001) | 0.242 |

Key: Data are n (%; row percentage) or median (IQR) for continuous variable and non-normal data in the descriptive table; CHR: Crude hazard ratio; aHR: adjusted for all other covariates in table; HR: hazard ratio; CI: Confidence Interval. [1]Time to ICU from TBH admission (in days); ICU: intensive care unit; HIV: human immunodeficiency virus; TB: tuberculosis; PaCO2: partial pressure of carbon dioxide; PaO2: partial pressure of oxygen; PF ratio: ratio of arterial oxygen partial pressure (in mmHg) to fractional inspired oxygen. Variables with p-value less than 0.15 in the unadjusted model, was kept in the final multivariable models and considered multicollinearity among predictors. As a result, we have excluded So2c in an adjusted model due to multicollinearity with pH even though p-value is <0.15 in an unadjusted model.

(95% CI 38–52%), with older patients having a higher mortality rate [84.4% (95% CI 83.3–85.4)] than younger patients [47.9% (95% CI 46.4–49.4%)] [29]. Our mortality rate was higher than the overall estimate in this systematic review [30], and the South African COVID-19 admission mortality rate was 26% during the study period [31]. The increased risk of in-hospital death associated with patients who were older was further augmented by the presence of one or more chronic comorbidities. There were significant gender differences in discharge

**Table 5. Medication use associated with hazard ratio among COVID-19 patients admitted in ICU.**

| Medications used | Hazard ratio | | | |
|---|---|---|---|---|
| | Crude HR | | Adjusted HR | |
| | CHR (95% CI) | p-value | aHR (95% CI) | p-value |
| Antibiotics | | | | |
| Co-amoxiclav | 1.17 (0.88–1.54) | 0.277 | | |
| Azithromycin | 1.28 (0.97–1.68) | 0.078 | 1.29 (0.94–1.78) | 0.113 |
| Meropenem | 0.74 (0.56–0.98) | 0.033 | 1.23 (0.89–1.71) | 0.205 |
| Vancomycin | 0.55 (0.41–0.75) | <0.001 | 0.61 (0.41–0.89) | 0.011 |
| Colistin | 0.49 (0.33–0.73) | <0.001 | 0.62 (0.38–1.01) | 0.053 |
| Corticosteroids | | | | |
| Dexamethasone | 0.90 (0.70–1.16) | 0.429 | | |
| Methyl prednisone | 1.32 (0.89–1.95) | 0.174 | | |
| Hydrocortisone | 0.95 (0.69–1.30) | 0.754 | | |
| Antifungal | | | | |
| Fluconazole | 0.58 (0.39–0.88) | 0.011 | 0.99 (0.62–1.59) | 0.963 |
| Anticoagulants | | | | |
| Prophylaxis enoxaparin | 0.14 (0.05–0.39) | <0.001 | 0.16 (0.06–0.46) | 0.001 |
| Other medications | | | | |
| Vitamin C | 1.23 (0.93–1.63) | 0.134 | 0.99 (0.71–1.37) | 0.933 |
| Thiamine | 1.28 (0.75–2.21) | 0.368 | | |
| Proton pump inhibitor | 0.61 (0.46–0.80) | <0.001 | 0.71 (0.53–0.97) | 0.033 |
| Aspirin | 0.81 (0.63–1.06) | 0.121 | 1.02 (0.77–1.36) | 0.867 |
| Anti-hypertensives | | | | |
| Spironolactone | 0.42 (0.26–0.69) | 0.001 | 0.50 (0.29–0.85) | 0.011 |
| Losartan | 0.51 (0.33–0.78) | 0.002 | 0.58 (0.36–0.95) | 0.030 |
| Simvastatin | 0.93 (0.67–1.28) | 0.652 | | |

**Table 6. Laboratory parameters associated with hazard ratio among COVID-19 patients admitted to ICU.**

| Laboratory parameters | Hazard ratio | | | |
|---|---|---|---|---|
| | Crude HR | | Adjusted HR[1] | |
| | CHR (95% CI) | p-value | aHR (95% CI) | p-value |
| Sodium | 1.01 (0.98–1.03) | 0.672 | | |
| Creatinine | 1.00 (1.000–1.001) | 0.214 | | |
| Haemoglobin | 1.00 (0.93–1.07) | 0.930 | | |
| Haematocrit | 1.75 (0.16–19.23) | 0.647 | | |
| Platelets | 1.00 (1.000–1.001) | 0.969 | | |
| Urea | 1.02 (1.00–1.04) | 0.048 | 1.01 (0.98–1.04) | 0.644 |
| White cell count | 1.00 (0.97–1.02) | 0.791 | | |
| Lymphocytes | 1.06 (0.86–1.31) | 0.558 | | |
| Neutrophils | 1.00 (0.97–1.03) | 0.954 | | |
| Chloride | 1.00 (0.98–1.03) | 0.706 | | |
| C-reactive protein | 1.00 (1.000–1.002) | 0.074 | 1.00 (1.00–1.001) | 0.823 |
| Procalcitonin | 1.00 (1.000–1.002) | 0.143 | 1.00 (1.00–1.002) | 0.148 |
| D-Dimer | 1.01 (0.99–1.03) | 0.371 | | |
| Ferritin | 1.00 (1.00–1.00) | 0.451 | | |
| INR | 1.19 (0.78–1.80) | 0.417 | | |
| aPTT | 1.03 (1.01–1.05) | 0.001 | | |
| NT-proBNP | 1.00 (1.00–1.00) | <0.001 | 1.00 (1.00–1.0001) | 0.039 |
| Troponin T | 1.00 (1.00–1.002) | 0.003 | 1.00 (0.997–0.002) | 0.976 |
| Total bilirubin | 1.01 (0.98–1.04) | 0.711 | | |
| Albumin | 0.99 (0.94–1.04) | 0.774 | | |
| ALT | 1.00 (1.00–1.001) | 0.317 | | |
| AST | 1.00 (1.00–1.001) | 0.567 | | |
| ALP | 1.00 (0.997–1.01) | 0.345 | | |
| Calcium | 0.40 (0.14–1.14) | 0.086 | 0.33 (0.11–1.02) | 0.055 |
| Magnesium | 1.93 (0.88–4.20) | 0.099 | 2.03 (0.89–4.65) | 0.092 |
| Phosphate | 1.13 (0.74–1.74) | 0.565 | | |
| HbA1c | 1.01 (0.95–1.07) | 0.771 | | |
| eGFR (non-normal) | 0.90 (0.69–1.19) | 0.461 | | |

Abbreviations: INR: international normalized ratio; aPTT, Activated Partial Thromboplastin Time; NT-proBNP: N-terminal (NT)-pro hormone BNP; ALT: alanine aminotransferase; AST: aspartate aminotransferase; ALP: Alkaline phosphatase; HbA1c: glycated haemoglobin; eGFR: estimated glomerular filtration rate.

[1]We excluded for total observation (n = Discharged + Death) < 250 in the adjusted analysis (e.g., aPTT)

rate (30% females versus 44% males) and mortality (70% females versus 56% males), on the CIRR, but not on the aIRR.

In multivariate Poisson regression analysis, seventeen categorical variables were identified as being associated with mortality among patients admitted to the ICU. Among these factors were: age, intubation, HIV positive status, arterial blood gas results, lactate, PF ratio, urea, neutrophil count, C-reactive protein (CRP), Procalcitonin (PCT), D-Dimer, NT-proBNP, troponin T, HbA1c, magnesium, aspartate transaminase (AST), and alkaline phosphatase (ALP). Among them, the age of ≥ 48 years, intubation, HIV status, and low pH all significantly predicted mortality after adjusting for potential confounders. Even though the above variables were not adjusted for all other covariates, the discussion below showed their association with COVID-19 mortality in different studies.

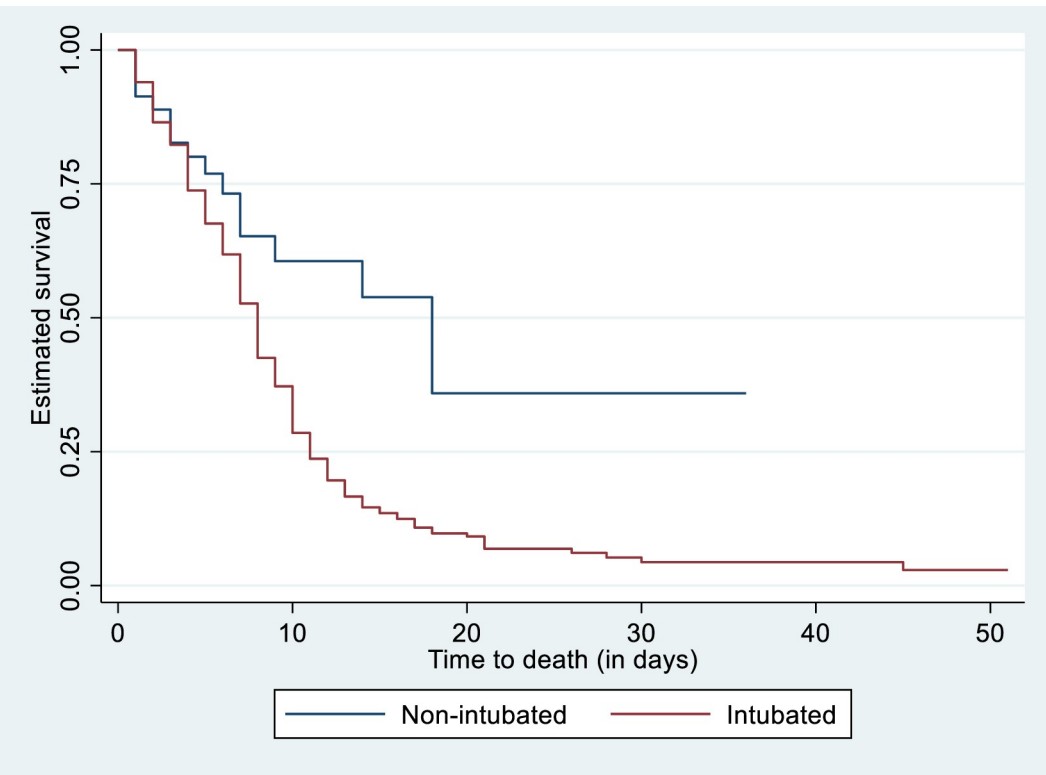

**Fig 2. Kaplan-Meier product limit estimates of survival among intubated and non-intubated COVID-19 patients admitted to ICU.**

Available evidence supports a significant association between older age and COVID-19 mortality in the ICU [11, 32]. Our study supported this observation with an adjusted IRR 1.01 (95% CI: 1.01, 1.02)). In the univariate analysis, HIV-positive status was associated with mortality among COVID-19 patients admitted to the ICU. This finding is similar to a systematic review and meta-analysis including 22 studies, and about 21 million participants showing that an HIV positive status was associated with a higher risk of mortality from COVID-19 [19, 33]. Several studies have found that the risk of ICU admission and death for COVID-19 among people living with HIV (PLWH) increased with age, consistent with an increased burden of comorbid conditions in older people [13, 34–37]. In contrast, another South African study found that while the proportion of PLWH was similar in surviving and deceased COVID-19 cases, a higher proportion of COVID-19 deaths occurred in patients aged 50 years or older in those living with HIV versus those who did not [38]. Diabetes (50%) and hypertension (42%) were present in a considerable proportion of PLWH who died from COVID-19 [38]. The median (IQR) age of 56.7 (48.0–63.1) years was found to be associated with COVID-19 mortality in our study. This is consistent with findings in other studies showing a preponderance for older people [34]. This study did not consider the crucial factors to consider among HIV individuals such as CD4 counts, viral loads, and ART use. In COVID-19 infection, a lower CD4 cell count in PLWH was associated with increased mortality [39]. In the same line, a mortality odds ratio of 2.85 (95% CI 1.26–6.44) for CD4 cell count less than 350 cells/ml compared to higher values [38, 40] and a higher mortality for CD4 cell count less than 200 cells/ml [39] were found. Similarly, the risk of COVID-19 hospitalization was 47% lower in people taking TDF/FTC (rate ratio 0.53, 95% CI 0.27–0.97) compared to people taking tenofovir alafenamide

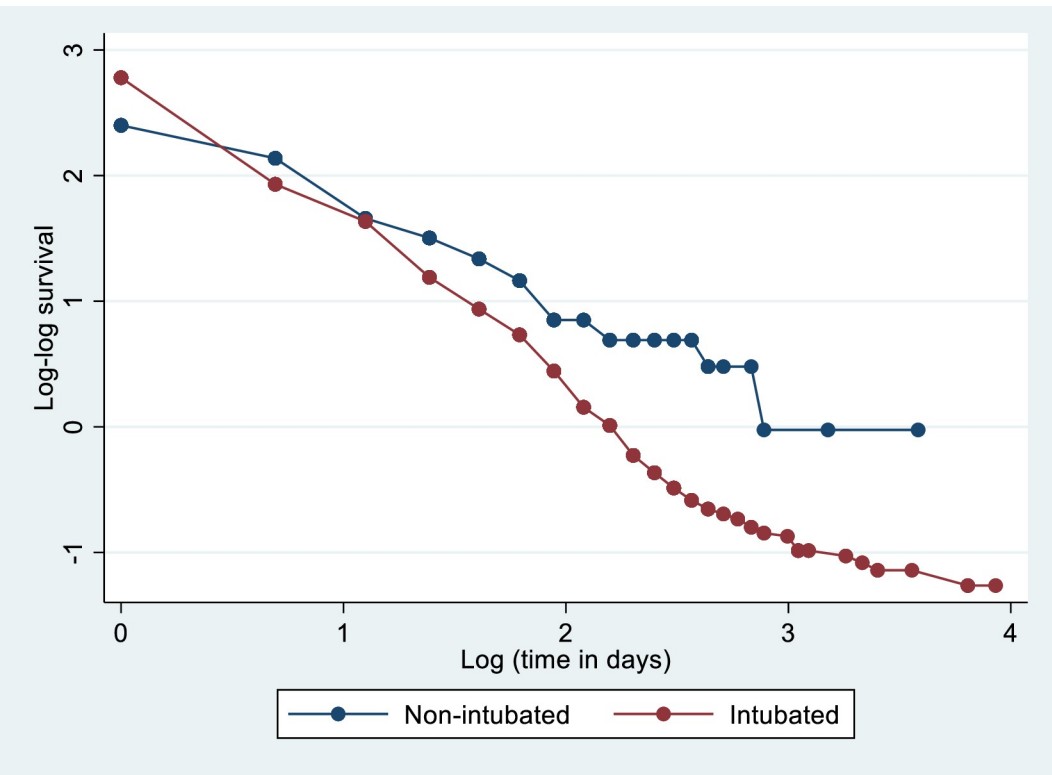

**Fig 3. Log-log plot of survival of COVID-19 patients admitted to ICU to assess the assumption of proportional hazards.**

(TAF)/FTC [35, 41]. In analyses adjusted for baseline renal function, the hazard ratio of COVID-19 death for TDF/FTC compared to abacavir (ABC) or zidovudine (AZT) was 0.41 (95% CI 0.21–0.78) in South Africa [38]. Given the low rate of PLWH on ART in this study, HIV as a risk factor for mortality should be considered with caution. An uncontrolled HIV viral load and/or a low CD4 count are directly related to a lack of recent or poor ART adherence. Tamuzi et al., demonstrated that SARS-CoV-2 could cause transient suppression of cellular immunity in PLWH, predisposing the patients to exacerbated reactivation or new TB infection [20]. However, our study did not collect data on active PTB in our study. Because only previous TB cases were reported, this could be explained by lack of PTB screening among COVID-19 ICU patients.

Regarding the blood gases and ventilation, our study found a significant association between arterial-blood gas parameters and the mortality rate. A $paO_2 < 6.9$ kPa, pH $< 7.40$ and $>7.49$ were associated with the CIRR of 0.96 (95%CI: 0.93–0.99) and 0.27 (95%CI: 0.17–0.43) respectively. Since 201/250 (80.4%) who died required IMV and had lower PF ratios, the severity of their ARDS was probably greater (although both groups would be graded as severe since both had their median PF ratios $< 100$). This may be explained by the fact that the decision to intubate was left to the discretion of the attending clinician and handled on a case-by-case basis. Furthermore, the admission PF ratio was significantly higher for those who survived intubation compared with those who died after being intubated 93.7 (IQR: 63.7–141.4) and 70.3 (51.0–91.25); respectively. This may imply that earlier intubation was advantageous, or that these patients were not as ill and therefore may have survived without intubation. After adjustment of the HR, the initial pH was associated with admission to the ICU. The CIRR of $PaCO_2$ higher than 4.9 was associated with a higher mortality rate. Our findings are

concordant with a study reporting a 10% increase in $FiO_2$ on the first day of ICU admission being associated with increased mortality (HR, 1.24; 95% CI, 1.20–1.27), whereas a 100-point increase in $PaO_2/FiO_2$ ratio decreased the hazard for mortality (HR, 0.66; 95%CI, 0.61–0.71) by 44% [11].

Our findings also indicated that being intubated and mechanically ventilated in the ICU was associated with a high mortality rate (aIRR 2.57, 95%CI: 2.08–3.18). A review of twelve non-randomized cohort studies found no statistically significant difference in all-cause mortality between patients undergoing early versus late intubation [30]. It is possible that there was an unrecognized superinfection with *aspergillosis*, *Acinetobacter baumannii*, *Klebsiella pneumonia*, and other bacteria among critically ill intubated COVID-19 patients, and that their presence may change the natural course of the disease [42, 43]. Furthermore, increasing the occupancy of beds compatible with mechanical ventilation is associated with a higher mortality risk for patients admitted to the ICU [44].

Furthermore, our results were supported by a meta-analysis that revealed that common markers that influenced patient outcomes were peripheral white blood cell values and acute phase reactants [45]. Interestingly, this review noted that the neutrophil count, lymphocyte count, CRP, and D-dimer levels showed trends on whether a patient would have a mild-to-moderate course of disease vs. a severe course or eventual death [45]. Regarding PCT, our results are in line with previous findings that higher PCT is associated with higher mortality in SARS-CoV-2 pneumonia as well as in critically ill patients in general [46–48].

Evidence of COVID-19-associated increases in circulating cardiac troponin T is emerging in the literature. The mechanism of SARS-CoV-2-induced cardiac injury is still unclear. The result of an autopsy study by Xu et al., demonstrated a few interstitial mononuclear inflammatory infiltrates in heart muscle, indicating inflammation [49]. A meta-analysis found patients to be at a high risk of death when troponin levels were elevated [50]. Further, troponin >13.5 ng/ml was associated with a greater chance of developing critical COVID-19 and ICU admission with adverse outcomes [51].

In our study, we found that an elevated AST level was significantly related to the mortality rate. These findings were supported by a recent study that noted a significant association between alkaline phosphate and critical COVID-19 illness and mortality [46]. Transaminases have been linked to poorer clinical outcomes such as respiratory failure, pneumonia, COVID-19 severity, and mortality, implying that these outcomes were likely related to liver injury [46].

Apart from COVID-19 being a risk factor for secondary bacterial and fungal nosocomial infections, published data suggest that combination antibiotic therapy may further predispose patients to these secondary infections [52, 53]. Furthermore, most of the pathogenic organisms found in COVID-19 patients are multidrug-resistant (MDR) nosocomial organisms. Finally, according to our findings, spironolactone was not associated with increased mortality. According to a recent nationwide case-control study, spironolactone did not affect the development of COVID-19 complications, [54]. In terms of other antihypertensive medications, two recent systematic reviews and meta-analyses found that angiotensin-converting enzyme inhibitors and angiotensin-receptor blockers do not appear to increase the risk of developing severe COVID-19 stages of the disease or mortality [55, 56].

The presence of established AKI was a major deciding factor for admission to the ICU. Only 11% developed AKI during their ICU stay. There were no differences in the proportion of patients that developed AKI between patients that died or were discharged. Patients were admitted to the ICU based on the sequential organ failure assessment score (SOFA score). Patients with established AKI at the time of needing ICU admission had high SOFA scores, which resulted in their exclusion. This may be the reason why AKI was not a predictor of mortality on regression analysis.

SARS-CoV-2 changes over time due to the mutation of different viral proteins. Some mutations may affect the virus's properties, such as how easily it spreads, the associated disease severity, or the performance of vaccines, therapeutic medicines, diagnostic tools, or other public health and social measures [57]. Those variants are named variants of concerns (VOCs). From the initial Wuhan variant, different VOCs have taken over South Africa among which Alpha, Beta, Delta, and omicron and their different lineages are predominant [57]. Compared with early variants of SARS-CoV-2 included in this study, new VOCs may be associated with an increase in the risk of ICU admission and death due to COVID-19.

The strength of this study is that it is a prospective cohort conducted in a critical care environment in Sub-Saharan Africa that gathered longitudinal data on various exposures including several co-morbidities, clinical parameters, demographic, hematologic, biochemical, and therapeutic factors that were assessed as potential risk factors for mortality among COVID-19. In addition, most of our results have been validated in previous meta-analysis and large cohort studies. Lastly, an in-depth multivariate analysis was conducted to adjust for possible confounders.

The determination of the mortality rate and associated risk factors among COVID-19 patients admitted to the ICU was a significant strength of our study. Our research has some limitations. The study is of observational nature and therefore no effect of any specific treatment or management strategy can be concluded. Furthermore, because this is a single-hospital design cohort and our criteria for admission to ICU may be different to others; our results may not be generalizable. The cut-offs for various haematological and biochemical risk factors of mortality in COVID-19 may also be debatable. Other limitations included CD4 counts, and viral load were not recorded among PLWH. Thus, we could not adjust for these parameters among PLWH. The study did not capture either the SOFA or the APACHE scores. Since the data were collected prior to COVID-19 vaccination, vaccination-related information was not included. Lastly, the number of comparisons made increases the likelihood of one or more spurious associations.

## 5. Conclusion

In comparison to other COVID-19 mortality studies conducted in ICU nationally and internationally, this study had a significantly higher mortality rate 65% (262/400). Our study demonstrated that advanced age, intubation, HIV positive status, low pH and PF ratio, high urea, lactate, neutrophil count, PCT, D-dimer, proBNP, troponin T, HbA1c, magnesium, high aspartate aminotransferase and alkaline phosphatase were all associated with mortality. After adjusting for potential confounders, age with median (IQR) of 56.7 (48.0–63.1) years, being intubated, HIV status, PCT, troponin, AST, and metabolic acidosis with median (IQR) of 7.46 (7.40–7.49) all significantly predicted mortality among hospitalized COVID-19 patients in the ICU. Our study supports previous findings regarding mortality risk factors in COVID-19 patients admitted to the ICU. A better understanding and identification of risk factors that may predispose to ICU admission may be required for more active medical decision-making to optimize patient outcomes. Our findings revealed a significant difference in demographic data, comorbidities, and laboratory characteristics that influenced mortality. Vulnerable populations are at increased risk for severe disease.

## Supporting information

**S1 Table. A: Frequency distribution of medication used among COVID-19 patients admitted in ICU.**
(DOCX)

**S2 Table. B: Frequency distribution of laboratory parameters at initial measurements among COVID-19 patients admitted in ICU.**
(DOCX)

## Acknowledgments

The authors would like to thank the Executive Management of the Faculty of Medicine and Health Sciences, Stellenbosch University, and the CEO of Tygerberg Hospital for supporting the COVID-19 Multidisciplinary Research Response Initiative. We acknowledge the support of IT Staff (Mr Wielligh Lambrechts) with technical assistance with development of the RED-Cap database data integration from multiple sources. Prof Peter S Nyasulu and Prof Brian Allwood take full responsibility for this work, which includes the study design, data acquisition and quality control, data access as well as the decision to submit and publish the manuscript in PLOS One. This work is dedicated to our colleague Prof Birhanu T. Ayele, the Lead Biostatistician of the Covid-19 Research Response team who passed on during the course of drafting this work. May his soul rest in eternal peace. We would also like to express our gratitude to the COVID-19 Research Response Collaboration at Stellenbosch University's Faculty of Medicine and Health Sciences.

## Author Contributions

**Conceptualization:** Peter S. Nyasulu, Birhanu T. Ayele, Coenraad F. Koegelenberg, Anteneh Yalew, Lovemore N. Sigwadhi, Brian W. Allwood.

**Data curation:** Peter S. Nyasulu, Coenraad F. Koegelenberg, Muhammad Saadiq Moolla, Anteneh Yalew, Nicola Baines, Padi Maud, Rory Dunbar, Lovemore N. Sigwadhi, Veranyuy D. Ngah, Brian W. Allwood.

**Formal analysis:** Birhanu T. Ayele, Anteneh Yalew, Lovemore N. Sigwadhi, Brian W. Allwood.

**Investigation:** Peter S. Nyasulu, Coenraad F. Koegelenberg, Elvis Irusen, Usha Lalla, Razeen Davids, Yazied Chothia, Francois Retief, Marianne Johnson, Stephen Venter, Renilda Pillay, Hans Prozesky, Jantjie Taljaard, Arifa Parker, Portia Jordan, Sa'ad Lahri, M Rafique Moosa, Muhammad Saadiq Moolla, Elizabeth Louw, Andre Nortje, Jacques L. Tamuzi, Annalise Zemlin, Zivanai Chapanduka, René English, Brian W. Allwood.

**Methodology:** Peter S. Nyasulu, Birhanu T. Ayele, Anteneh Yalew, Lovemore N. Sigwadhi, Jacques L. Tamuzi.

**Project administration:** Peter S. Nyasulu, Nicola Baines, Padi Maud, Annalise Zemlin, René English.

**Resources:** Peter S. Nyasulu.

**Supervision:** Peter S. Nyasulu, Coenraad F. Koegelenberg, Elvis Irusen, Eric H. Decloedt, Veranyuy D. Ngah, Zivanai Chapanduka, Brian W. Allwood.

**Validation:** Peter S. Nyasulu, Rory Dunbar.

**Writing – original draft:** Peter S. Nyasulu, Birhanu T. Ayele, Coenraad F. Koegelenberg, Elvis Irusen, Anteneh Yalew, Lovemore N. Sigwadhi, Veranyuy D. Ngah, Jacques L. Tamuzi, Brian W. Allwood.

**Writing – review & editing:** Peter S. Nyasulu, Birhanu T. Ayele, Coenraad F. Koegelenberg, Elvis Irusen, Usha Lalla, Razeen Davids, Yazied Chothia, Francois Retief, Marianne Johnson, Stephen Venter, Renilda Pillay, Hans Prozesky, Jantjie Taljaard, Arifa Parker, Eric H. Decloedt, Portia Jordan, Sa'ad Lahri, M Rafique Moosa, Muhammad Saadiq Moolla, Anteneh Yalew, Nicola Baines, Padi Maud, Elizabeth Louw, Andre Nortje, Rory Dunbar, Lovemore N. Sigwadhi, Veranyuy D. Ngah, Jacques L. Tamuzi, Annalise Zemlin, Zivanai Chapanduka, René English, Brian W. Allwood.

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
