## [Decision Letter · Decision Letter 0]

8 Mar 2022

PONE-D-21-17355Clinical characteristics associated with mortality of COVID-19 patients admitted to an Intensive Care Unit of a tertiary hospital in South AfricaPLOS ONE

Dear Dr. Nyasulu,

Thank you for submitting your manuscript to PLOS ONE. After careful consideration, we feel that it has merit but does not fully meet PLOS ONE’s publication criteria as it currently stands. Therefore, we invite you to submit a revised version of the manuscript that addresses the points raised during the review process. The reviewers raised a number of concerns regarding your study. They felt that too much of the discussion was devoted to associations that were not shown to be statistically significant. They also felt that aspects of the methodology/study design such as patient inclusion criteria, whether any patients had active TB, etc. were not presented. Their comments can be viewed in full, below.

We look forward to receiving your revised manuscript.

Kind regards,

Natasha McDonald, PhD

Associate Editor

PLOS ONE

Journal Requirements:

Reviewers' comments:

Reviewer's Responses to Questions

**Comments to the Author**

1. Is the manuscript technically sound, and do the data support the conclusions?

Reviewer #1: Yes

Reviewer #2: Partly

2. Has the statistical analysis been performed appropriately and rigorously? 

Reviewer #1: Yes

Reviewer #2: Yes

3. Have the authors made all data underlying the findings in their manuscript fully available?

Reviewer #1: Yes

Reviewer #2: Yes

4. Is the manuscript presented in an intelligible fashion and written in standard English?

Reviewer #1: Yes

Reviewer #2: Yes

5. Review Comments to the Author

Reviewer #1: In this study Nyasulu et al describe Clinical characteristics associated with mortality of COVID-19 patients admitted to an

Intensive Care Unit of a tertiary hospital in South Africa. The authors conclude that the mortality rate in COVID-19 patients

admitted to the ICU was high and that older age, the need for invasive mechanical ventilation,HIV status, and metabolic acidosis were significant predictors of mortality. Overall the manuscript is of potential interest. My comments are as follows:

1. My major comment is that there is to much focus on discussing possible associations that did not remain significant in adjusted analysis. The authors should revise results and discussion appropriately focusing on main results.

2. Results, page 16 ".....In the adjusted multivariate Cox regression analysis indicated that intubated patients

in ICU were 1.56 times at higher risk of death than those who were not intubated (aHR

1.56, 95%CI: 1.08-2.25, P = 0.017). Similarly, HIV-infected patients had 1.64 times

increased risk of dying compared HIV negative patients (aHR 1.64, 95%CI: 1.11-2.36,

P = 0.012). In contrast, myalgia on admission was associated with 32% reduced risk

of dying compared to those who did not present with this clinical symptom (aHR 0.68,

95%CI: 0.47-0.97, P = 0.032) [Table 3a]." This sentence is repeated on page 17

3. same for statement " Furthermore, use of vancomycin,

enoxaparin, proton pump inhibitors (PPI), spironolactone, losartan and other

hypertensive drugs were associated with a low risk of mortality (aHR 0.62, 95%CI:

0.38-1.01, P = 0.011; aHR = 0.16, 95%CI: 0.06-0.46, P = 0.001; aHR 0.71, 95%CI:

0.53-0.97, P = 0.033; aHR = 0.50, 95%CI: 0.29-0.85, P = 0.011; aHR 0.58, 95%CI:

0.36-095, P = 0.03 and aHR 0.74, 95%CI: 0.55-0.99, P = 0.04 respectively) [Table 3b]." that is repeated on page 18

It would be good to see all drugs that were examined to have possible association in table 3

4. It would be useful to report actual values of CRP, NT proBNP, CBC and not only HR

5. Where there any data on complications such as VAP, pulmonary embolism or pneumotorax?

6. For Cox regression model-what was the follow up time? Fig 3 should be clearly labeled

7. What was the criteria for admission to ICU? Were there any patients on NIV? Was HFO available at the general ward/ high dependency unit as well?

8. Mortality of intubated patients is likely related to severity of illness, or complications as authors noted, but time of intubation cannot be commented as it was not investigated.

9. Please consider citing Risk Factors for Coronavirus Disease 2019 (COVID-19) Death in a Population Cohort Study from the Western Cape Province, South Africa. Clin Infect Dis. 2021 Oct 5;73(7):e2005-e2015. doi: 10.1093/cid/ciaa1198

Minor there is no need top repeat p values in discussion

Reviewer #2: The authors have presented important clinical data on COVID-19-related mortality in a large cohort of people (n=402) admitted to the intensive care unit (ICU) of a large teaching hospital in Cape Town, South Africa prior to COVID-19 vaccine availability. The authors provide important insights into factors contributing to COVID-19-related deaths in a middle-income setting with a relatively younger population and high burden of HIV compared to countries reporting similar data in the early days of the pandemic. Of particular interest is the fact that although people living with HIV (PLHIV) make up a small proportion of people requiring ICU admission for COVID-19 (14%), HIV remained a strong predictor of mortality in multivariable analysis 1.30 (1.06, 1.59).

General comments:

While the authors highlight the importance of the finding that HIV was a significant predictor of morality amongst ICU patients, the finding seems to get lost in the discussion owing to a lot of comparative data on crude results that are not generally in need of explanation.

As a reader, I am curious to know if any measures of CD4 count or viral load were taken or if at least an indication of ART-use (or is this a study limitation) to provide more insight into the relationship of HIV and COVID-19 mortality. Moreover, it would be important to highlight that although the common comorbidities leading to severe disease in high-income countries (diabetes, hypertension, and other cardiovascular diseases) were also common amongst patients admitted to the ICU in the present setting (despite being less prevalent in the general public), they were not significantly predictive of mortality when considered in the model (although asthma and increased BMI may be worth noting as trending toward significance). It would also be interesting to note (perhaps in an appendix) what comorbidities were most common among PLHIV in the cohort.

It would also be helpful to note whether or not any patients admitted had active TB disease. In line 227 on page 10, the authors report 7% of people admitted had TB, but then refer to it as a “history of previous TB” in Table 1. If no one had active TB in ICU, this would be worth highlighting, although might there be a risk that concomitant TB went undetected given the focus on severe COVID-19 disease?

Specific comments:

Abstract:

On page 3, lines 57-60, the authors mention sub-Saharan Africa’s lower COVID-19 case count, and potential explanations for it, but do not adequately come back to this point in the abstract’s conclusion or the manuscript’s discussion. I would suggest either excluding it from the abstract (and background) or returning to it in the discussion/conclusion based on what was observed in the study and bringing in NICD data on excess deaths in Western Cape province – it seems likely that case counts were better reported in Western Cape in this period, so under-reporting may be at play in other settings but less so in this one, and that lower prevalence of non-communicable diseases may also contribute.

Page 3, line 74-75 : “after adjusting for incident rate ratios” should read “after adjusting for confounding”

Methods

Page 9, line 192, the authors mention socioeconomic status as a sociodemographic variable. It would be helpful to know how this was measured (if indeed it was) and to report results by SES in tables 1 and 2.

It is not clear why age was dichotomized at the median, as age groups have been typically reported in other studies. Does the model still hold if age is dichotomized at >65 or as a continuous variable?

Please include a definition of low versus high pH under methods. This is important to be able to note if the statistically significant difference is also clinically significant.

Findings:

Please review tables and legends closely, as there are places where numbers and asterisks are not explained or comments between authors were included. It would also help to have clear cut points indicated for clinical parameters and arterial blood gasses in Table 3a. I believe from the interpretation of HRs that these are hazard ratios, not hazard rates.

Please note that the paragraph immediately preceding Table 3a is a repeat from page 16 (no line numbers are visible here), as are the paragraphs immediately proceeding Tabe 3a and 3b.

Discussion:

It would help the reader to focus the discussion on the predictive factors of mortality and their clinical significance. As mentioned above, HIV is an important finding that needs more context.

While it may seem obvious, it is also important to note that patients placed on mechanical ventilation were more likely sicker than those who were not, and it is unlikely that the use of mechanical ventilation led to mortality.

The authors should also note under limitations that the number of comparisons made increases the likelihood of one or more spurious associations (we would expect 1/20 to be spurious with a significance of p=0.05).

6. PLOS authors have the option to publish the peer review history of their article (what does this mean?). If published, this will include your full peer review and any attached files.

Reviewer #1: **Yes: **Marija Vukoja

Reviewer #2: No

---

## [Author Response · Author response to Decision Letter 0]

28 May 2022

Responses to reviewers’ feedback 24th May 2022 

Thank you for sending us the comprehensive feedback. We are grateful to you for this wonderful feedback, it has helped us to improve the quality of the manuscript and turn it a more informative piece of work for the readers. The manuscript entitled: “Clinical characteristics associated with mortality of COVID-19 patients admitted to an Intensive Care Unit of a tertiary hospital in South Africa” has been revised accordingly. 

Reviewer #1: 

In this study Nyasulu et al describe Clinical characteristics associated with mortality of COVID-19 patients admitted to an Intensive Care Unit of a tertiary hospital in South Africa. The authors conclude that the mortality rate in COVID-19 patients

admitted to the ICU was high and that older age, the need for invasive mechanical ventilation, HIV status, and metabolic acidosis were significant predictors of mortality. Overall, the manuscript is of potential interest. My comments are as follows:

Comment 1: My major comment is that there is too much focus on discussing possible associations that did not remain significant in adjusted analysis. The authors should revise results and discussion appropriately focusing on main results.

Response 1: Many thanks, we agree completely, and the discussion has been strengthened on page 27-28, lines 407-437. We have removed non statistically adjusted analysis in lines 478-484, 520-523, 528-535, 544-569) original version).

Comment 2: Results, page 16 ".....In the adjusted multivariate Cox regression analysis indicated that intubated patients in ICU were 1.56 times at higher risk of death than those who were not intubated (aHR 1.56, 95%CI: 1.08-2.25, P = 0.017). Similarly, HIV-infected patients had 1.64 times increased risk of dying compared HIV negative patients (aHR 1.64, 95%CI: 1.11-2.36, P = 0.012). In contrast, myalgia on admission was associated with 32% reduced risk of dying compared to those who did not present with this clinical symptom (aHR 0.68, 95%CI: 0.47-0.97, P = 0.032) [Table 3a]." This sentence is repeated on page 17

Response 2: Thanks for this observation. On page 24, lines 363-369 (original version), this paragraph has been removed.

Comment 3: same for statement " Furthermore, use of vancomycin,

enoxaparin, proton pump inhibitors (PPI), spironolactone, losartan and other

hypertensive drugs were associated with a low risk of mortality (aHR 0.62, 95%CI:

0.38-1.01, P = 0.011; aHR = 0.16, 95%CI: 0.06-0.46, P = 0.001; aHR 0.71, 95%CI:

0.53-0.97, P = 0.033; aHR = 0.50, 95%CI: 0.29-0.85, P = 0.011; aHR 0.58, 95%CI:

0.36-095, P = 0.03 and aHR 0.74, 95%CI: 0.55-0.99, P = 0.04 respectively) [Table 3b]." that is repeated on page 18

It would be good to see all drugs that were examined to have possible association in table 3

Response 3: Thanks for this comment. Amendment has been done accordingly. This amendment is on Page 25, lines 380-385. This paragraph has been removed (original document).

Comment 4: It would be useful to report actual values of CRP, NT proBNP, CBC and not only HR

Response 4: Thanks. Page 18-19, Line 298-303, we have added actual values of CRP, NT proBNP, CBC in Table 2b.

Comment 5: Were there any data on complications such as VAP, pulmonary embolism or pneumotorax?

Response 5: Thanks for this valuable concerns. On Page 31, lines 509-514, we have highlighted this point. “We did not record cases related to Ventilator-Associated Pneumonia, Pulmonary Embolism or Pneumothorax in our data. Many patients died suddenly, and as post-mortems were not performed on COVID-19 patients at the time, the cause of death (including suspected pulmonary embolism) could never be confirmed. This is unfortunately a limitation of this study.

Comment 6: For Cox regression model-what was the follow up time? Fig 3 should be clearly labelled.

Response 6: Many thanks for the comment. Page 22-23, the follow up time was in days with longest patient staying more than 50 days in ICU. This has been corrected in Fig 2 and which has now been clearly labelled.

Comment 7: What was the criteria for admission to ICU? Were there any patients on NIV? Was HFO available at the general ward/ high dependency unit as well?

Response 7: Thanks for identifying our missing point. During the first wave NIV and HFNO were not available in general wards and the high dependency unit was incorporated in ICU. The admission criteria and triage document are summarised in reference 28 and was strictly adhered to. On Page 8, line 167-170. We have added the following: “The initial assessment of the referred patient is focused on determining whether the patient is critically ill and requires ICU admission for ventilatory support or other organ support that is only available in the ICU [28].”.

Comment 8: Mortality of intubated patients is likely related to severity of illness, or complications as authors noted, but time of intubation cannot be commented as it was not investigated.

Response 8: Thanks. On page 29, line 415-418, we have removed this statement (Original document).

Comment 9: Please consider citing Risk Factors for Coronavirus Disease 2019 (COVID-19) Death in a Population Cohort Study from the Western Cape Province, South Africa. Clin Infect Dis. 2021 Oct 5;73(7):e2005-e2015. doi: 10.1093/cid/ciaa1198

Response 9: Thanks. this reference has been included in the discussion on the line 461-463.

 

Reviewer #2: 

The authors have presented important clinical data on COVID-19-related mortality in a large cohort of people (n=402) admitted to the intensive care unit (ICU) of a large teaching hospital in Cape Town, South Africa prior to COVID-19 vaccine availability. The authors provide important insights into factors contributing to COVID-19-related deaths in a middle-income setting with a relatively younger population and high burden of HIV compared to countries reporting similar data in the early days of the pandemic. Of particular interest is the fact that although people living with HIV (PLHIV) make up a small proportion of people requiring ICU admission for COVID-19 (14%), HIV remained a strong predictor of mortality in multivariable analysis 1.30 (1.06, 1.59).

General comments:

While the authors highlight the importance of the finding that HIV was a significant predictor of morality amongst ICU patients, the finding seems to get lost in the discussion owing to a lot of comparative data on crude results that are not generally in need of explanation.

As a reader, I am curious to know if any measures of CD4 count or viral load were taken or if at least an indication of ART-use (or is this a study limitation) to provide more insight into the relationship of HIV and COVID-19 mortality. Moreover, it would be important to highlight that although the common comorbidities leading to severe disease in high-income countries (diabetes, hypertension, and other cardiovascular diseases) were also common amongst patients admitted to the ICU in the present setting (despite being less prevalent in the general public), they were not significantly predictive of mortality when considered in the model (although asthma and increased BMI may be worth noting as trending toward significance). It would also be interesting to note (perhaps in an appendix) what comorbidities were most common among PLHIV in the cohort. It would also be helpful to note whether or not any patients admitted had active TB disease. In line 227 on page 10, the authors report 7% of people admitted had TB, but then refer to it as a “history of previous TB” in Table 1. If no one had active TB in ICU, this would be worth highlighting, although might there be a risk that concomitant TB went undetected given the focus on severe COVID-19 disease?

Response: Thank for this concern. Page 27-28 , line 407-435. We have clarified this concern as follows: “Several studies have found that the risk of ICU admission and death for COVID-19 among people living with HIV (PLWH) increased with age, consistent with an increased burden of comorbid conditions in older people [15, 35-38]. In contrast, another South African study found that while the proportion of PLWH was similar in surviving and deceased COVID-19 cases, a higher proportion of COVID-19 deaths occurred in patients aged 50 years or older in those living with HIV versus those who did not [39]. Diabetes (50%) and hypertension (42%) were present in a significant proportion of PLWH who died from COVID-19 [39]. The median (IQR) age of 56.7 (48.0-63.1) years was found to be associated with COVID-19 mortality in our study. This is consistent with findings in older people [35]. This study did not take into account important HIV confounders such as CD4 counts, viral load, and ART use. In COVID-19 infection, a lower CD4 cell count among PLWH was associated with increased mortality [40]. In the same line, a mortality odds ratio of 2.85 (95 percent CI 1.26–6.44) for CD4 cell count less than 350 cells/ml compared to higher values [39, 41] and higher mortality for CD4 cell count less than 200 cells/ml [39] were found. Similarly, the risk of COVID-19 hospitalization was 47 percent lower in people taking tenofovir disoproxil fumarate (TDF)/emtricitabine (FTC) (rate ratio 0.53, 95 percent CI 0.27–0.97) compared to people taking tenofovir alafenamide (TAF)/FTC [35, 42]. In analyses adjusted for baseline renal function, the hazard ratio of COVID-19 death for TDF/FTC compared to abacavir (ABC) or zidovudine (AZT) was 0.41 (95 percent CI 0.21–0.78) in South Africa [39]. An uncontrolled HIV viral load and/or a low CD4 count are directly related to a lack of recent or poor ART adherence. Tamuzi et al. demonstrated that SARS-CoV-2 could cause transient suppression of cellular immunity in PLWH, predisposing patients to exacerbated reactivation or new TB infection [23]. However, we did not collect data on active PTB in our study. Because only previous TB cases were reported, this could be explained by poor PTB screening among COVID-19 ICU patients”.

Specific comments:

Abstract:

Comment 1: On page 3, lines 57-60, the authors mention sub-Saharan Africa’s lower COVID-19 case count, and potential explanations for it, but do not adequately come back to this point in the abstract’s conclusion or the manuscript’s discussion. I would suggest either excluding it from the abstract (and background) or returning to it in the discussion/conclusion based on what was observed in the study and bringing in NICD data on excess deaths in Western Cape province – it seems likely that case counts were better reported in Western Cape in this period, so under-reporting may be at play in other settings but less so in this one, and that lower prevalence of non-communicable diseases may also contribute.

Response 1: Thanks for this constructive comment. Line 57-60, 112-118. We have removed this paragraph from the abstract and background (original document).

Comment 2: Page 3, line 74-75: “after adjusting for incident rate ratios” should read “after adjusting for confounding”

Response 2: Thanks, this has been corrected. Page 3, line 74. This is now read “after adjusting for confounding”.

Methods

Comment 3: Page 9, line 192, the authors mention socioeconomic status as a sociodemographic variable. It would be helpful to know how this was measured (if indeed it was) and to report results by SES in tables 1 and 2.

It is not clear why age was dichotomized at the median, as age groups have been typically reported in other studies. Does the model still hold if age is dichotomized at >65 or as a continuous variable?

Response 3: Many thanks for the comment. Line 191-192, we have corrected as demographic and lifestyle characteristics. These include age, sex, smoking status and alcohol use.

Comment 4: Please include a definition of low versus high pH under methods. This is important to be able to note if the statistically significant difference is also clinically significant.

Response 4: Many thanks, we agree completely, and information has been added on Page 9, line 195: “A pH above 7.45 was considered as an alkalemia”.

Findings:

Comment 5: Please review tables and legends closely, as there are places where numbers and asterisks are not explained or comments between authors were included. It would also help to have clear cut points indicated for clinical parameters and arterial blood gasses in Table 3a. I believe from the interpretation of HRs that these are hazard ratios, not hazard rates.

Response 5: Thanks for this observation. This is well noted. We have made the change in Table 1, Table 2a, 2b, and 3a. This is seen in the lines 243-342 

Comment 6: Please note that the paragraph immediately preceding Table 3a is a repeat from page 16 (no line numbers are visible here), as are the paragraphs immediately preceding Table 3a and 3b.

Response 6: Thanks for the comment. Page 24-25, line 386-391, this paragraph has been removed. (Original document)

Discussion:

Comment 7: It would help the reader to focus the discussion on the predictive factors of mortality and their clinical significance. As mentioned above, HIV is an important finding that needs more context.

Response 7: Thanks for the observation. We have broadly resolved this concern through the earlier response and adjustment made on. Page 27-28, line 407-435 in the general comment response.

Comment 8: While it may seem obvious, it is also important to note that patients placed on mechanical ventilation were more likely sicker than those who were not, and it is unlikely that the use of mechanical ventilation led to mortality.

Response 8: Thanks for this comment. This Section has been added as per recommendation. Page 29, line 461-463: “Furthermore, increasing the occupancy of beds compatible with mechanical ventilation is associated with a higher mortality risk for ICU patients [45].”.

Comment 9: The authors should also note under limitations that the number of comparisons made increases the likelihood of one or more spurious associations (we would expect 1/20 to be spurious with a significance of p=0.05).

Response 9: Thanks. Page 37, line 518-522. We have added the following: “the number of comparisons made increases the likelihood of one or more spurious associations”.

---

## [Decision Letter · Decision Letter 1]

22 Jul 2022

PONE-D-21-17355R1Clinical characteristics associated with mortality of COVID-19 patients admitted to an Intensive Care Unit of a tertiary hospital in South AfricaPLOS ONE

Dear Dr. Nyasulu,

Thank you for submitting your manuscript to PLOS ONE. After careful consideration, we feel that it has merit but does not fully meet PLOS ONE’s publication criteria as it currently stands. Therefore, we invite you to submit a revised version of the manuscript that addresses the points raised during the review process.

We look forward to receiving your revised manuscript.

Kind regards,

Alexandru Rogobete, MD, PhD, MSc, ClinRes

Academic Editor

PLOS ONE

Reviewers' comments:

Reviewer's Responses to Questions

**Comments to the Author**

1. If the authors have adequately addressed your comments raised in a previous round of review and you feel that this manuscript is now acceptable for publication, you may indicate that here to bypass the “Comments to the Author” section, enter your conflict of interest statement in the “Confidential to Editor” section, and submit your "Accept" recommendation.

Reviewer #1: (No Response)

Reviewer #3: (No Response)

2. Is the manuscript technically sound, and do the data support the conclusions?

Reviewer #1: Yes

Reviewer #3: Partly

3. Has the statistical analysis been performed appropriately and rigorously? 

Reviewer #1: Yes

Reviewer #3: No

4. Have the authors made all data underlying the findings in their manuscript fully available?

Reviewer #1: Yes

Reviewer #3: (No Response)

5. Is the manuscript presented in an intelligible fashion and written in standard English?

Reviewer #1: Yes

Reviewer #3: No

6. Review Comments to the Author

Reviewer #1: The authors have significantly improved the manuscript. However, the discussion should be more focused on specificity of COVID 19 outcomes in South Africa, and not focus to much on variables that were not associated in multivariate analysis. Also, regarding statistical analysis it should be clearly stated which variables were included into final model.

Reviewer #3: In this perspective, cohort study the authors reported the experience of ICU admission in patients with COVID-19 in a South African hospital, with the primary aim to assess the survival of these patients.

I understand this is a second-round revision, however, in my humble opinion, the manuscript continues to suffer major flaws that need to be considered by investigators. I agree with the previous reviewer who suggested applying more attention to the high prevalence of HIV infection in the South African population, which could give this manuscript uniqueness. However, the authors did not represent their own HIV patients with COVID-19. I believe the author needs to get more information about their own HIV patients.

As a reader of a manuscript coming from South Africa, I’ll be more interested to know about the South African variant of COVID-19. Unfortunately, the investigators did not address this critical issue in the current manuscript.

U Lalla et al reported the experience of South African hospital in ICU patients with COVID-19 (Afr J Thorac Crit Care Med 2021 Dec 31;27(4):10.7196/AJTCCM.2021.v27i4.185). The authors need to streamline why the current manuscript needs to be published as soon as there is a reported experience in critical ill COVID-19 patients from the same country.

No assessment of disease severity was provided. There is neither SOFA score nor APACHE score reported in the Tables. This is mandated in any study assessing outcomes of patients in ICU.

The manuscript had many writing errors in grammar and punctuation that need revision, as well as flaws that need to be considered by the authors:

Introduction

Line#108 the author reported many complications of severe COVID-19, among these, was encephalitis. In the cited reference [12] encephalitis was not among the reported complications. Please revise and consider adding VTE development from another reference.

Methodology

The planned recording of co-morbidities for the study did not include chronic renal failure, which appeared later in the result section. Please clarify in the methodology section all the recorded chronic illnesses.

Include either SOFA or APACHE score.

Line #192, the primary outcome of interest (of interest) appears to be redundant.

This is a prospective study. Please determine if the study patients were consented prior to enrollment, or if the investigators granted a consent waiver by the ethical body that cleared the study. This information needs to be addressed clearly in the methodology section.

Results

Tuberculosis could be abbreviated as TB as it appeared in the introduction section.

In the result section, line#231 the word days was unnecessarily repeated. This was observed in different parts of the result section. Please revise.

In Table.1 what was the aim of reporting the sample sizes related to each clinical characteristic? This need to be described for the reader. I suggest revising Table 1 in a way to facilitate reader understanding.

In line #245 the author reported the oxygenation in writing (less than 90) while it was reported as ( < ) at others. Please unify.

How was the PF ratio calculated in patients on admission to the hospital? Did all the patients undergo arterial line assessment on admission to the hospital? Some patients might be admitted with moderate COVID-19 and progress to severe during their hospital stay. Which PF ratio was entered for analysis?

How was the admission PF ratio statistically compared for those who died and those who survived? No statistical test was provided in the methodology section to perform head-to-head comparisons. The same applies to the comparison between those who were intubated in ICU versus before ICU admission (line#268). Please add to the statistical part section.

Table.2 For myalgia and DBP on univariate analysis show P value of 0.086. No P value was reported after adjustment ( P > 0.15). The same applies to variables reported under arterial blood pressure.

Many medications were selected for association testing. Which was the base to select these medications? (Table 3b). In the discussion section, anti-retroviral agents used to treat HIV were discussed without introducing this agent for association with COVID-19 outcomes. I believe as soon as this is a study in the area of the world with a high prevalence of AIDS, more focus needs to be directed toward medications used to manage HIV infection and to present them with analysis. The authors need to report in detail dexamethasone use with doses and duration.

In Table 3c the author reported eGFR. What was the method for assessing the eGFR. Nothing was mentioned in the methodology section.

In Table 3b, for the anticoagulants (enoxaparin) was that a therapeutic or prophylactic dose? This is crucial to report to know how many patients with indications for full anticoagulants were enrolled.

Discussion

The author continued to report the P values of the study in the discussion section. This should be transferred to the results section. ( for example lines 373 up to 375 on the first page of the discussion).

In the discussion section,

Line#376 extremely high is an overstatement. Please revise

Line#379, missing punctuation, and revising for the (% sign).

The study suffers many limitations. The authors need to add more of limitations in this study.

7. PLOS authors have the option to publish the peer review history of their article (what does this mean?). If published, this will include your full peer review and any attached files.

Reviewer #1: **Yes: **Marija Vukoja MD PhD

Reviewer #3: No

---

## [Author Response · Author response to Decision Letter 1]

10 Sep 2022

Manuscript: Clinical characteristics associated with mortality of COVID-19 patients admitted to an Intensive Care Unit of a tertiary hospital in South Africa 

Comment 1: In this perspective, cohort study the authors reported the experience of ICU admission in patients with COVID-19 in a South African hospital, with the primary aim to assess the survival of these patients. I understand this is a second-round revision, however, in my humble opinion, the manuscript continues to suffer major flaws that need to be considered by investigators. I agree with the previous reviewer who suggested applying more attention to the high prevalence of HIV infection in the South African population, which could give this manuscript uniqueness. However, the authors did not represent their own HIV patients with COVID-19. I believe the author needs to get more information about their own HIV patients. 

Response 1: Thank you for this valuable comment. In the manuscript with track changes line 300-302, we have included more information about the treatment of HIV-infected patients. “Among HIV-infected patients, only (13.9%) 54/388 were on ART and tenofovir disoproxil fumarate (TDF)/emtricitabine (FTC)/efavirenz (EFV) was the prevalent regimen (43.54%)”.

Comment 2: As a reader of a manuscript coming from South Africa, I’ll be more interested to know about the South African variant of COVID-19. Unfortunately, the investigators did not address this critical issue in the current manuscript. 

Response 2: Thanks for this comment. In the manuscript with track changes line 523-531, we have included the following: “SARS-CoV-2 changes over time due to the mutation in different viral proteins. Some mutations may affect the virus’s properties, such as how easily it spreads, the associated disease severity, or the performance of vaccines, therapeutic medicines, diagnostic tools, or other public health and social measures [55]. Those variants are named variants of concerns (VOCs). From the initial Wuhan variant, different VOCs have taken over South Africa, Alpha, Beta, Delta, and C.1.2 omicron and its different lineages are predominant in South Africa [55]. Compared with early variants of SARS-CoV-2 included in this study, new VOCs may be associated with an increase in the risk of ICU admission and death due to COVID-19. 

Comment 3: U Lalla et al reported the experience of South African hospital in ICU patients with COVID-19 (Afr J Thorac Crit Care Med 2021 Dec 31;27(4):10.7196/AJTCCM.2021.v27i4.185). The authors need to streamline why the current manuscript needs to be published as soon as there is a reported experience in critical ill COVID-19 patients from the same country. Lalla et al. compared the clinical characteristics, management, and outcomes of COVID-19 patients admitted to a South African ICU during the first and second waves. However, this manuscript focuses primarily on the first wave. In comparison to Lalla et al., this paper describes all epidemiological and clinical aspects of the first wave. This manuscript is significant because it was written at a time when there were few studies on COVID-19 patients admitted to the ICU in South Africa.

Response 3: Thank you for the valuable comment.

Comment 4: No assessment of disease severity was provided. There is neither SOFA score nor APACHE score reported in the Tables. This is mandated in any study assessing outcomes of patients in ICU. 

Response 4: Thanks for this comment, we did not gather data on the SOFA score or the APACHE score at the pick of the COVID-19 as case records did not have documented these scores.pa. This has been listed as the study’s limitation in the track change manuscript line 559-560.

Comment 5: The manuscript had many writing errors in grammar and punctuation that need revision, as well as flaws that need to be considered by the authors: 

Response 5: Thanks for this valuable comment. We have revised the grammar, punctuation, and flow throughout the manuscript.

Introduction 

Comment 6: Line#108 the author reported many complications of severe COVID-19, among these, was encephalitis. In the cited reference [12] encephalitis was not among the reported complications. Please revise and consider adding VTE development from another reference. 

Response 6: Thanks. In the manuscript with track changes line 111, the reference related to encephalitis has been highlighted. 

Methodology: 

Comment 7: The planned recording of co-morbidities for the study did not include chronic renal failure, which appeared later in the result section. Please clarify in the methodology section all the recorded chronic illnesses. 

Response 7: Thanks. In the manuscript with track changes line 195, we have included: “chronic kidney disease (CKD)”.

Comment 8: Include either SOFA or APACHE score. 

Response 8: Thanks for this comment, we did not capture either the SOFA score or the APACHE score in our study. This has been listed as the study’s limitation line 559-560.

Comment 9: Line #192, the primary outcome of interest (of interest) appears to be redundant.

Response 9: Thanks. We have corrected this in the manuscript with track, lines 198-199.

Comment 10: This is a prospective study. Please determine if the study patients were consented prior to enrollment, or if the investigators granted a consent waiver by the ethical body that cleared the study. This information needs to be addressed clearly in the methodology section. 

Response 10: Thanks. In the manuscript with track changes line 151, we have included the following: “The ethical body that approved the study granted the investigators a consent waiver”.

Results

Comment 11: Tuberculosis could be abbreviated as TB as it appeared in the introduction section. 

Response 11: Thanks. “Tuberculosis” has been abbreviated as “TB” in the manuscript with track changes line 237.

Comment 12: In the result section, line#231 the word days was unnecessarily repeated. This was observed in different parts of the result section. Please revise.

Response 12: Thanks. In the manuscript with track changes line 240, 274 the word “days” has been deleted.

Comment 13: In Table.1 what was the aim of reporting the sample sizes related to each clinical characteristic? This need to be described for the reader. I suggest revising Table 1 in a way to facilitate reader understanding. 

Response 13: Many thanks for the comment. The purpose of presenting the sample size was to show the number of observations for each characteristic since each characteristic did not have an equal sample size. We have included this statement at the bottom of Table 1. Furthermore, we have combined the last two columns into one column in Table 1 (Tracked manuscript pages 13 and 14).

Comment 14: In line #245 the author reported the oxygenation in writing (less than 90) while it was reported as ( < ) at others. Please unify. 

Response 14: Thanks. In the manuscript track changes line 265, we have harmonized this concern.

Comment 15: How was the PF ratio calculated in patients on admission to the hospital? Did all the patients undergo arterial line assessment on admission to the hospital? Some patients might be admitted with moderate COVID-19 and progress to severe during their hospital stay. Which PF ratio was entered for analysis? 

Response 15: Thanks for this valuable comment. In this manuscript, we have considered the initial PF at admission. All the patients admitted to the ICU were severe to critical COVID-19 cases.

Comment 16: How was the admission PF ratio statistically compared for those who died and those who survived? No statistical test was provided in the methodology section to perform head-to-head comparisons. The same applies to the comparison between those who were intubated in ICU versus before ICU admission (line#268). Please add to the statistical part section. 

Response 16: Thank you for the comment. We have updated the statistical methods section. 

Chi-square and Wilcoxon ranksum tests were performed to test the population distribution associated with mortality among categorical variables and difference in medians for continuous variable with p-values. This statement has been added in statistical analysis section (Tracked manuscript lines 206-209).

As a result, the association of characteristics and the outcome are presented in Table 2a. In the current version, p-value is added for tests of association and equality of median (for example, admission PF ratio statistically compared for those who died and those who survived; the comparison between those who were intubated in ICU versus before ICU admission). Similarly, we have updated Table 2b (added p-value*: testing median equality of laboratory parameters compared with those who died and those who survived).

Comment 17: Table.2 For myalgia and DBP on univariate analysis show P-value of 0.086. No P value was reported after adjustment (P > 0.15). The same applies to variables reported under arterial blood pressure. 

Response 17: Thank you for the valuable comment. We have added all the characteristics that show p-value <0.15 in the unadjusted model in the final adjusted model in Table 2 (Tracked manuscript pages 16-18). As a result, Table 2a is updated in the manuscript (sore throat, myalgia, BP – diastolic also added in the adjusted model). However, there were correlations (multicollinearity issue) of pH with paCO2, K+, Lactate, HCO3std, So2c, and duration of pre-ICU admission; pf ratio with pao2, lactate, and so2c. As a result, we have excluded paCO2, paO2, Lactate, So2c, and duration of pre-ICU admission in an adjusted model due to multicollinearity even though the p-value is <0.15 in an unadjusted model. The above statement is added in Table 2a footnote in the manuscript. Table 3a is also updated in the manuscript since we have excluded So2c due to multicollinearity with pH even though the p-value is <0.15 in an unadjusted model. All results were also updated accordingly in the manuscript.

The summary of scatter matrix, correlation with a p-value of arterial blood gases parameter multicollinearity test is given below. 

Variables with the number of observations less than 250 (rule of thumb) were not included in the adjusted model (e.g., AST in Table 2b).

Comment 18: Many medications were selected for association testing. Which was the base to select these medications? (Table 3b). In the discussion section, anti-retroviral agents used to treat HIV were discussed without introducing this agent for association with COVID-19 outcomes. I believe as soon as this is a study in the area of the world with a high prevalence of AIDS, more focus needs to be directed toward medications used to manage HIV infection and to present them with analysis. The authors need to report in detail dexamethasone use with doses and duration. 

Response 18: Thank you for this comment. In the manuscript with track changes page 27, we had only the medications listed and added the corticosteroids in Table 3b and line 346-349. In the tracked manuscript line 300-302, we have included the following the rate of ART among PLWH: “However, among HIV-infected patients, only (13.9%) 54/388 were on ART and TDF/FTC/EFV was the main regimen (43.54%)”.

Comment 19: In Table 3c the author reported eGFR. What was the method for assessing the eGFR. Nothing was mentioned in the methodology section. 

Response 19: Thanks. In the manuscript with track changes line 195, we have added in the methodology section (in outcomes and predictor variables section) as a normal eGFR is 60 or more.

Comment 20: In Table 3b, for the anticoagulants (enoxaparin) was that a therapeutic or prophylactic dose? This is crucial to report to know how many patients with indications for full anticoagulants were enrolled. 

Response 20: Thanks for this comment. In table 3b, page 27, we have specified that enoxaparin was given as a prophylactic dose. This means that enoxaparin was administered to all the COVID-19 patients admitted to the ICU.

Discussion 

Comment 21: The author continued to report the P values of the study in the discussion section. This should be transferred to the results section (for example lines 373 up to 375 on the first page of the discussion). 

Response 21: Thanks. In the manuscript with track changes lines 405, 419, 432, 434, 467, 468, 475, 479, 481 and 483, we have corrected this.

Comment 22: Line#376 extremely high is an overstatement. Please revise 

Response 22: Thanks. In the manuscript with track changes, line 408, we have removed the adverb “extremely” 

Comment 23: Line#379, missing punctuation, and revising for the (% sign). 

Response 23: Thanks. In the manuscript with track changes lines 450, 453, and 455, the change has been made.

Comment 24: The study suffers many limitations. The authors need to add more limitations to this study.

Response 24: Thanks. In the manuscript with track changes, lines 559-560, we have included the following: “The study did not capture either the SOFA or the APACHE scores”.

---

## [Decision Letter · Decision Letter 2]

6 Nov 2022

PONE-D-21-17355R2Clinical characteristics associated with mortality of COVID-19 patients admitted to an Intensive Care Unit of a tertiary hospital in South AfricaPLOS ONE

Dear Dr. Nyasulu,

Thank you for submitting your manuscript to PLOS ONE. After careful consideration, we feel that it has merit but does not fully meet PLOS ONE’s publication criteria as it currently stands. Therefore, we invite you to submit a revised version of the manuscript that addresses the points raised during the review process.

We look forward to receiving your revised manuscript.

Kind regards,

Hani Amir Aouissi, Ph.D.

Academic Editor

PLOS ONE

Journal Requirements:

Reviewers' comments:

Reviewer's Responses to Questions

**Comments to the Author**

1. If the authors have adequately addressed your comments raised in a previous round of review and you feel that this manuscript is now acceptable for publication, you may indicate that here to bypass the “Comments to the Author” section, enter your conflict of interest statement in the “Confidential to Editor” section, and submit your "Accept" recommendation.

Reviewer #4: (No Response)

2. Is the manuscript technically sound, and do the data support the conclusions?

Reviewer #4: Yes

3. Has the statistical analysis been performed appropriately and rigorously? 

Reviewer #4: Yes

4. Have the authors made all data underlying the findings in their manuscript fully available?

Reviewer #4: (No Response)

5. Is the manuscript presented in an intelligible fashion and written in standard English?

Reviewer #4: Yes

6. Review Comments to the Author

Reviewer #4: I reviewed the paper PONE-D-21-17355R2 by Nyasulu et al. submitted for publication in PLOS One.

The paper is very interesting and pleasant to read. Given the fact that it’s a second revised version I’m only going to ask for minor revisions. Here are some comments: (I am using the line numbers of R2 version).

1. Abstract, Line 70 (Of 402 patients…etc) please rephrase.

2. Line 95, I would like you to add at least a sentence specifying for example that susceptibility to SARS-CoV-2 is universal, but older age have always been associated with disease severity (and high mortality), add this citation to support your statement:

*https://doi.org/10.3390/healthcare10071341

3. Line 96, please update data.

4. Line 112, COVID-19 instead of Covid-19

5. Line 142, please add rheumatic diseases with diabetes and hypertension and add these citations:

*https://doi.org/10.1001/jama.2020.12839

*https://doi.org/10.1016/j.nmni.2021.100846

6. Line 164, COVID-19 again, please correct it in the whole document.

7. Your results section is much more improved compared to the first submission, congratulations for that.

8. The fact that this study was conducted in 2020, there was no vaccines at that moment, this can be considered as a limitation that should be added in your limitations section.

9. I suggest adding more results (numbers) in your conclusion.

10. I also suggest to the authors to make another English revision, I know that it’s the second revised version, but it’s better to carefully check the manuscript again.

Good luck.

7. PLOS authors have the option to publish the peer review history of their article (what does this mean?). If published, this will include your full peer review and any attached files.

Reviewer #4: **Yes: **mostefa ababsa

---

## [Author Response · Author response to Decision Letter 2]

8 Dec 2022

Reviewers ‘responses

Manuscript: Clinical characteristics associated with mortality of COVID-19 patients admitted to an Intensive Care Unit of a tertiary hospital in South Africa 

Comment 1: Abstract, Line 70 (Of 402 patients…etc) please rephrase.

Response 1: Thanks. In the tracking manuscript line 71-73, we have rephrased as follows: “Of the 402 patients admitted to the ICU, 250 (62%) died, and another 12 (3%) died in the hospital after being discharged from the ICU”.

Comment 2: Line 95, I would like you to add at least a sentence specifying for example that susceptibility to SARS-CoV-2 is universal, but older age have always been associated with disease severity (and high mortality), add this citation to support your statement:

*https://doi.org/10.3390/healthcare10071341

Response 2: Thank you for this substantial input. In the tracking manuscript, line 96-98, we have included this: “Although SARS-CoV-2 susceptibility is universal, older age has always been associated with disease severity (and high mortality) [4]”.

Comment 3: Line 96, please update data.

Response 3: Thank you for making this observation. We have updated the COVID-19 data in the tracking manuscript lines 99-101 to include data from November 7, 2022.

Comment 4: Line 112, COVID-19 instead of Covid-19

Response 4: Thanks. We have corrected this as highlighted in the tracking manuscript line 115

Comment 5: Line 142, please add rheumatic diseases with diabetes and hypertension and add these citations:

*https://doi.org/10.1001/jama.2020.12839

*https://doi.org/10.1016/j.nmni.2021.100846

Response 5: Thanks. These two references are included in the tracking manuscript on line 147. It now reads: “however, there is little known about clinical outcomes of COVID-19 patients with HIV, TB, and post-TB lung disease (PTLD), rheumatic diseases, diabetes and hypertension [26, 27]”.

Comment 6: Line 164, COVID-19 again, please correct it in the whole document.

Response 6: Thank you; this has been corrected in tracking manuscript line 167.

Comment 7: Your results section is much more improved compared to the first submission, congratulations for that.

Response 7: Thank you so much for this appreciation.

Comment 8: The fact that this study was conducted in 2020, there were no vaccines at that moment, this can be considered as a limitation that should be added in your limitations section.

Response 8: Thanks for this. In the tracking manuscript line 557-559, we have included the following: “Since the data were collected prior to COVID-19 vaccination, vaccination-related information was not included”.

Comment 9: I suggest adding more results (numbers) in your conclusion.

Response 9: Thanks. We included the results as numbers in the conclusion of the tracking manuscript lines 564, 568-570.

Comment 10: I also suggest to the authors to make another English revision, I know that it’s the second revised version, but it’s better to carefully check the manuscript again.

Response 10: Thank you for your helpful feedback; we have revised the English throughout the manuscript.

---

## [Decision Letter · Decision Letter 3]

12 Dec 2022

Clinical characteristics associated with mortality of COVID-19 patients admitted to an Intensive Care Unit of a tertiary hospital in South Africa

PONE-D-21-17355R3

Dear Dr. Nyasulu,

We’re pleased to inform you that your manuscript has been judged scientifically suitable for publication and will be formally accepted for publication once it meets all outstanding technical requirements.

Kind regards,

Hani Amir Aouissi, Ph.D.

Academic Editor

PLOS ONE

---

## [Editor Report · Acceptance letter]

19 Dec 2022

PONE-D-21-17355R3 

Clinical characteristics associated with mortality of COVID-19 patients admitted to an Intensive Care Unit of a tertiary hospital in South Africa 

Dear Dr. Nyasulu:

I'm pleased to inform you that your manuscript has been deemed suitable for publication in PLOS ONE. Congratulations! Your manuscript is now with our production department. 

Kind regards, 

on behalf of

Dr. Hani Amir Aouissi 

Academic Editor

PLOS ONE